# Recurrent activating mutations of PPARγ associated with luminal bladder tumors

Natacha Rochel [1], Clémentine Krucker[2,3], Laure Coutos-Thévenot[2,3], Judit Osz[1], Ruiyun Zhang[2,3,9], Elodie Guyon[2,3], Wayne Zita[1], Séverin Vanthong[1], Oscar Alba Hernandez[4], Maxime Bourguet[4], Kays Al Badawy[1], Florent Dufour[2,3], Carole Peluso-Iltis[1], Syrine Heckler-Beji[1], Annick Dejaegere[1], Aurélie Kamoun [5], Aurélien de Reyniès[5], Yann Neuzillet [2,3], Sandra Rebouissou[2,3,10], Claire Béraud[6], Hervé Lang[7], Thierry Massfelder[8], Yves Allory[2,3], Sarah Cianférani[4], Roland H. Stote[1], François Radvanyi[2,3] & Isabelle Bernard-Pierrot[2,3]

The upregulation of PPARγ/RXRα transcriptional activity has emerged as a key event in luminal bladder tumors. It renders tumor cell growth PPARγ-dependent and modulates the tumor microenvironment to favor escape from immuno-surveillance. The activation of the pathway has been linked to PPARG gains/amplifications resulting in PPARγ overexpression and to recurrent activating point mutations of RXRα. Here, we report recurrent mutations of PPARγ that also activate the PPARγ/RXRα pathway, conferring PPARγ-dependency and supporting a crucial role of PPARγ in luminal bladder cancer. These mutations are found throughout the protein—including N-terminal, DNA-binding and ligand-binding domains— and most of them enhance protein activity. Structure-function studies of PPARγ variants with mutations in the ligand-binding domain allow identifying structural elements that underpin their gain-of-function. Our study reveals genomic alterations of PPARG that lead to pro-tumorigenic PPARγ/RXRα pathway activation in luminal bladder tumors and may open the way towards alternative options for treatment.

[1] Institut de Génétique et de Biologie Moléculaire et Cellulaire (IGBMC), Institut National de La Santé et de La Recherche Médicale (INSERM), U1258/Centre National de Recherche Scientifique (CNRS),  UMR7104/Université de Strasbourg, 67404 Illkirch, France. [2] Institut Curie, PSL Research University, CNRS, UMR144, Equipe Labellisée Ligue contre le Cancer, 75005 Paris, France. [3] Sorbonne Universités, UPMC Université Paris 06, CNRS, UMR144, 75005 Paris, France. [4] Laboratoire de Spectrométrie de Masse BioOrganique, Université de Strasbourg, CNRS,  IPHC UMR 7178, 67000 Strasbourg, France. [5] Ligue Nationale Contre le Cancer, Programme Cartes d'Identité des Tumeurs (CIT), 75013 Paris, France. [6] UROLEAD SAS, School of Medicine, 67085 Strasbourg, France. [7] Department of Urology, Nouvel Hôpital Civil Hôpitaux Universitaires de Strasbourg, Hôpitaux Universitaires de Strasbourg, 67091 Strasbourg, France. [8] INSERM UMRS1113, Section of Cell Signalization and Communication in Kidney and Prostate Cancer, INSERM and University of Strasbourg, School of Medicine, Fédération de Médecine Translationnelle de Strasbourg (FMTS), 67085 Strasbourg, France. [9] Present address: Department of Urology, Ren Ji Hospital, School of Medicine, Shanghai Jiao Tong University, Shanghai, China. [10] Present address: INSERM, UMR-1162 "Génomique Fonctionnelle des tumeurs solides", 75010 Paris, France. These authors contributed equally: Clémentine Krucker, Laure Coutos-Thévenot.  Correspondence and requests for materials should be addressed to N.R. (email: rochel@igbmc.fr) or to I.B-P. (email: isabelle.bernard-pierrot@curie.fr)

PARγ (peroxisome proliferator-activated receptor gamma) is a transcription factor of the nuclear receptor family that functions predominantly as a heterodimer with RXRα. PPARγ is a key regulator of glucose homeostasis and adipogenesis[1,2]. In addition to its well established role in adipocyte differentiation, it has also been shown to be involved in differentiation in several tissues including the urothelium[3]. Its role in cancer is far less understood. Historically, *PPARG* was considered to be a tumor suppressor[4], but several studies showed that it displays pro-tumorigenic effects in neuroblastoma, metastatic prostate cancer, and bladder carcinoma[4–8].

In bladder cancer, the fourth most frequent cancer in men in industrialized countries, *PPARG* focal amplifications, resulting in PPARγ and PPARγ target genes overexpression were observed in 12–17% of the muscle-invasive bladder carcinomas (MIBC) and in 10% of the non-muscle-invasive bladder carcinomas (NMIBCs)[7,9]. These copy number alterations of *PPARG* are associated with luminal tumors, a subtype of bladder tumors accounting for 75% of non-muscle-invasive bladder carcinomas (NMIBCs)[9,10] and 60% of muscle-invasive bladder carcinomas (MIBCs)[11]. The luminal MIBCs have been shown to display a PPARγ activation signature[7,12]. We previously demonstrated that PPARγ overexpression induces an oncogenic addiction in these tumors by showing that the loss of PPARγ expression inhibited bladder cancer cell viability[7], most notably observed in cell lines presenting *PPARG* gain or amplification[6,8]. Recurrent mutations of RXRα (S242F/Y) have also been identified in 5% of MIBCs and the luminal subgroup of MIBC tumors is enriched in these mutations[9,13]. These RXRα mutations are gain-of-function mutations that promote ligand-independent activation of PPARγ signaling pathway[13,14]. They drive proliferation of urothelial organoids in a tumor suppressor loss context[14], render bladder tumor cell growth PPARγ-dependent[14] and promote immune evasion in MIBCs[11]. Hyper-activation of PPARγ signaling, either due to *PPARG* gene amplification or an RXRα hotspot mutation (S427 F/Y), can be pharmacologically inhibited with PPARγ-selective inverse agonists that decrease the viability of PPARγ-dependent bladder cancer cells. This highlights PPARγ as a therapeutic target in luminal bladder tumors[8].

Given this crucial role of the PPARγ/RXRα pathway in bladder tumors, in this work, we search for other genetic alterations that could drive its activation in both NMIBC and MIBC. By sequencing PPARγ and RXRα in 359 tumors and studying publicly available data for 455 MIBC, we identify eight recurrent mutations of PPARγ associated with luminal tumors. Functional analyses reveal that five of these mutations enhance the transcriptional activity of PPARγ and that the activation of PPARγ pathway confers a PPAR-dependence to the cells. Biochemical analyses show that PPARγ mutations favor the recruitment of coregulators. Finally, by a structure-function analysis of three mutations affecting the ligand-binding domain of PPARγ, we demonstrate that these mutations promote the adoption of PPARγ active state accounting for the recruitment of coactivators. Our study provides additional genetic evidence for a pro-tumorigenic role of PPARγ in bladder cancer and strengthens the importance of the PPARγ/RXRα pathway in luminal bladder cancer.

## Results

### Recurrent mutations of PPARγ and RXRα in bladder tumors.
Using the conventional Sanger's method, we sequenced the *PPARG* and *RXRA exons* in 359 bladder tumors (199 of which were NMIBCs) from our CIT series of tumors (carte d'identité des tumeurs) and from a bank of samples collected at Strasbourg hospital, and in 25 bladder cell lines (Supplementary Table 1). We

detected *PPARG* mutations in 3.9% of the tumors (14/359) and in one cell line, and *RXRA* mutations in 1.7% of the tumors (6/338), all of which being MIBC, and in one cell line (Fig. 1a and Supplementary Tables 2 and 3). We also analyzed publicly available deep sequencing data from The Cancer Genome Atlas[9] genomic databases (405 MIBC samples)[10] (http://cancergenome.nih.gov) and from the Dana Farber & Memorial Sloan Kettering (MSKCC) cohort[15] (50 MIBC samples) (Supplementary Table 1). We detected a comparable *PPARG* mutation rate, 3.1% of the tumors (14/455), but a higher *RXRA* mutation rate, 6.1% of the tumors (28/455) (Fig. 1a and Supplementary Tables 2 and 3) in these tumor series compared to the previous tumor series analyzed. This higher mutation rate was mostly due to the absence of *RXRA* mutation in NMIBC in the CIT series of tumors. *PPARG* mutations were not mutually exclusive with either *RXRA* mutations ($p = 1$ and $p = 0.2105$ for conventional and next-generation sequencing, respectively) or *PPARG* amplification ($p = 1$ and $p = 1$ for conventional and next-generation sequencing, respectively), as shown by Fisher's exact test analyses. Two mutations (E3K and D7N) were specific to the PPARγ2 isoform (NM_015869) (Supplementary Table 2). We therefore numbered all mutations relative to this isoform, which is 28 amino acids longer than the PPARγ1 isoform (NM_138712) at the N-terminal end. The PPARγ2 isoform is expressed principally in adipocyte cells, as well as in urothelial cells, whereas PPARγ1 is ubiquitously expressed. As we aimed to identify genetic alterations that could activate the PPARγ/RXRα pathway, we focused on recurrent mutations that were likely to be gain-of-function mutations. Of the 21 unique PPARγ mutations identified here in bladder tumors, 6 were recurrent and occurred in both MIBC and NMIBC (Fig. 1b and Supplementary Table 2). Using publicly available data from the COSMIC (http://cancer.sanger.ac.uk/cosmic) or cBioPortal (http://www.cbioportal.org/) databases, we determined whether the 21 PPARγ mutations identified here were also present in other bladder tumors or in other tumor types. From this analysis, we found a seventh recurrent mutation (E3K) in bladder tumors (Fig. 1b and Supplementary Tables 2 and 4). Interestingly, four of these seven recurrent mutations and one unique mutation (M280I) identified in bladder tumors were also reported in other type of tumors in these databases, strengthening their likelihood to be activating mutations (Fig. 1b and Supplementary Table 4). We therefore considered in total of eight recurrent mutations of PPARγ for further studies (Fig. 1b, c). These recurrent PPARγ mutations ($p = 0.015$) and *PPARG* amplifications ($p = 1.5 \times 10^{-5}$) were significantly enriched in bladder tumors presenting a PPARγ activation signature, which were mostly luminal tumors ($p = 2.2 \times 10^{-16}$), as shown by Fisher's exact test. These mutations affect different major functional domains of the PPARγ protein: the N-terminal domain (E3K) including the area around the EGFR phosphorylation sites (S112) (P113S), the DNA-binding domain (R164W, R168K), the ligand-binding domain (S249L, T475M) including the area spatially close to the CDK5 phosphorylation site, S273 [(M280I, I290M)] (Fig. 1b, c). Visualization of recurrent mutations in the three-dimensional (3D) structure of the PPARγ/RXRα heterodimer revealed that the most frequent recurrent mutation of PPARγ(T475M) and the hotspot mutation of RXRα (S427Y/F) affected residues that co-localized spatially at the PPARγ/RXRα dimer interface, suggesting that both these mutations would affect heterodimer formation (Fig. 1c). Accordingly, RXRα (S427Y/F) mutations have been shown to enhance RXRα interaction with PPARγ and activation of the PPARγ/RXRα pathway[13].

### Recurrent mutations of PPARγ are gain-of-function.
We investigated the functional impact of seven of the eight recurrent mutations of PPARγ identified on the transcriptional activity

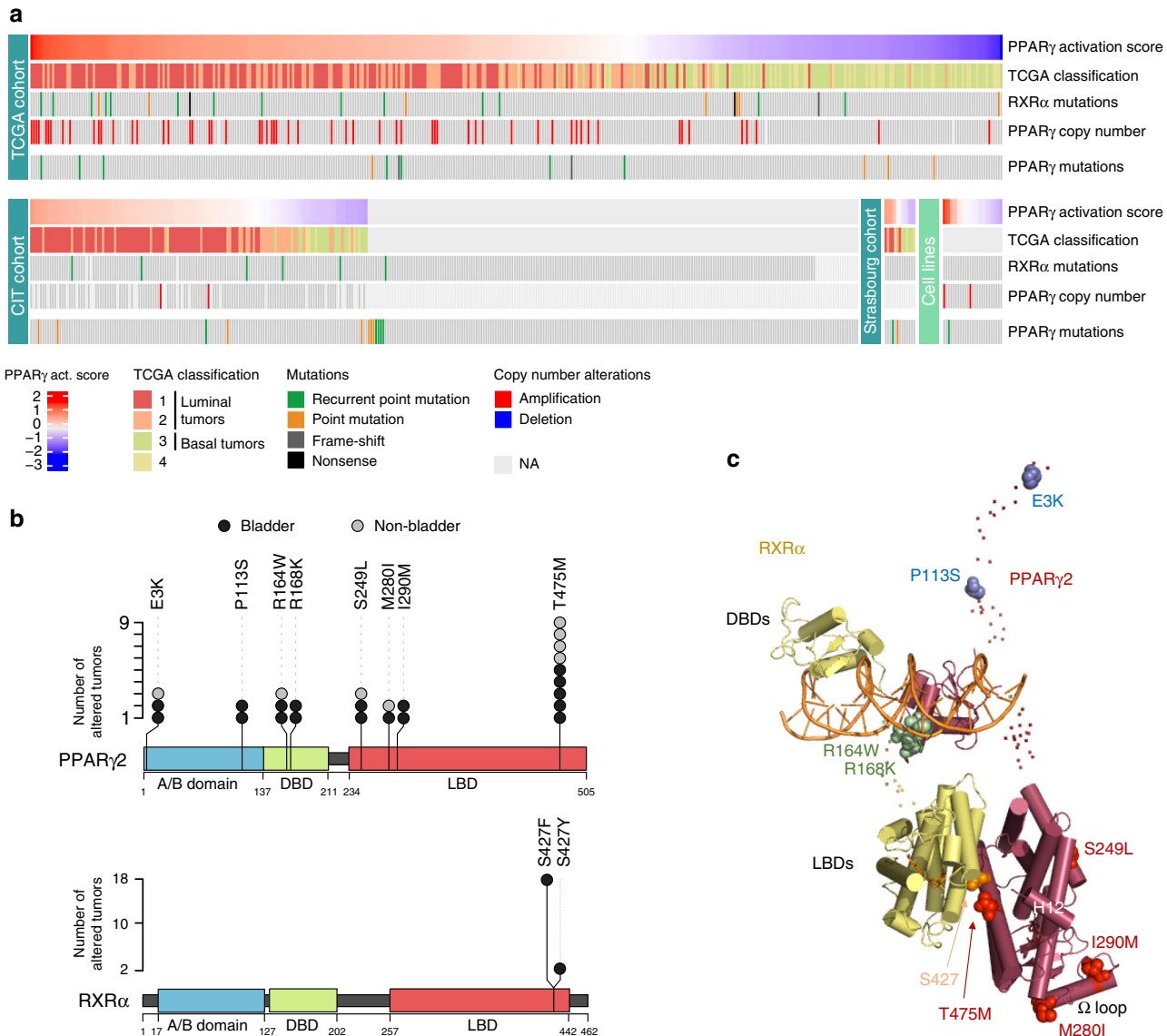

**Fig. 1** Mutations of *PPARγ* and *RXRα* in bladder tumors. **a** Oncoprints of *PPARγ* and *RXRα* mutations in three series of bladder tumors and a panel of 25 bladder tumor cell lines. Samples were sorted by PPARγ activation score when transcriptomic data were available. Source data are provided as a Source Data file. **b** Lolliplot representations of recurrent mutations of *PPARγ* (upper panel) and *RXRα* (lower panel) identified in the 859 bladder tumors studied here (black circles) or in publicly available data available for other bladder tumors and other types of cancer from the COSMIC and cBioPortal databases (Supplementary Table 4) (gray circles). Sequences are numbered according to the PPARγ2 isoform. A/B: N-terminal domain; DBD: DNA-binding domain; LBD: ligand-binding domain (LBD). **c** Position of the residues affected by the recurrent *PPARγ* and *RXRα* mutations on the full-length PPARγ-RXRα-DNA-coactivator peptide solution structure[51]. The folded domains are shown in cartoon representation and the disordered hinges and NTDs are shown as dots. The mutated residues are shown as spheres. The residues mutated in PPARγ are colored in blue (A/B domain), green (DBD), and red (LBD). The residue mutated in RXRα is colored in orange. Source data are provided as a Source Data file

of the protein in HEK293FT cells, using a luciferase reporter gene containing three copies of the DR1 sequence of *PPARγ* DNA response element (PPRE) arranged in tandem and linked to the thymidine kinase promoter (PPRE-3×-TK)[16]. The PPARγ P113S, R168K, S249L, M280I, I290M, and T475M mutant proteins had significantly higher levels of transcriptional activity than the wild type (WT) in the absence of exogenous ligand (two to six times higher), whereas the R164W mutant, which is located in the DNA-binding domain, had levels of activity similar to that of the WT (Fig. 2a). Similar results were obtained in the presence of 1 µM rosiglitazone, a synthetic PPARγ ligand (Supplementary Fig. 1A). Consistent with these findings, overexpression of the PPARγ mutants P113S, S249L, M280I, I290M,

and T475M, in 5637 cells, in the absence of exogenous PPARγ ligand, significantly enhanced the expression of several known PPARγ target genes (*FABP4*, *ACSL5*, and *PLIN2*) relative to the WT, as shown by RT-qPCR (Fig. 2b and Supplementary Fig. 1B, C). However, overproduction of both the R164W and R168W mutant proteins had an impact on PPARγ target gene expression similar to that of the WT protein (Fig. 2b). Similar results were obtained in presence of 500 nM rosiglitazone (Supplementary Fig. 1D). By combining the results of these two different approaches to measure PPARγ transcriptional activity, we clearly showed that five of the seven recurrent *PPARγ* mutations analyzed were gain-of-function mutations. The remaining two mutations, R164W and R168K, had no clear apparent

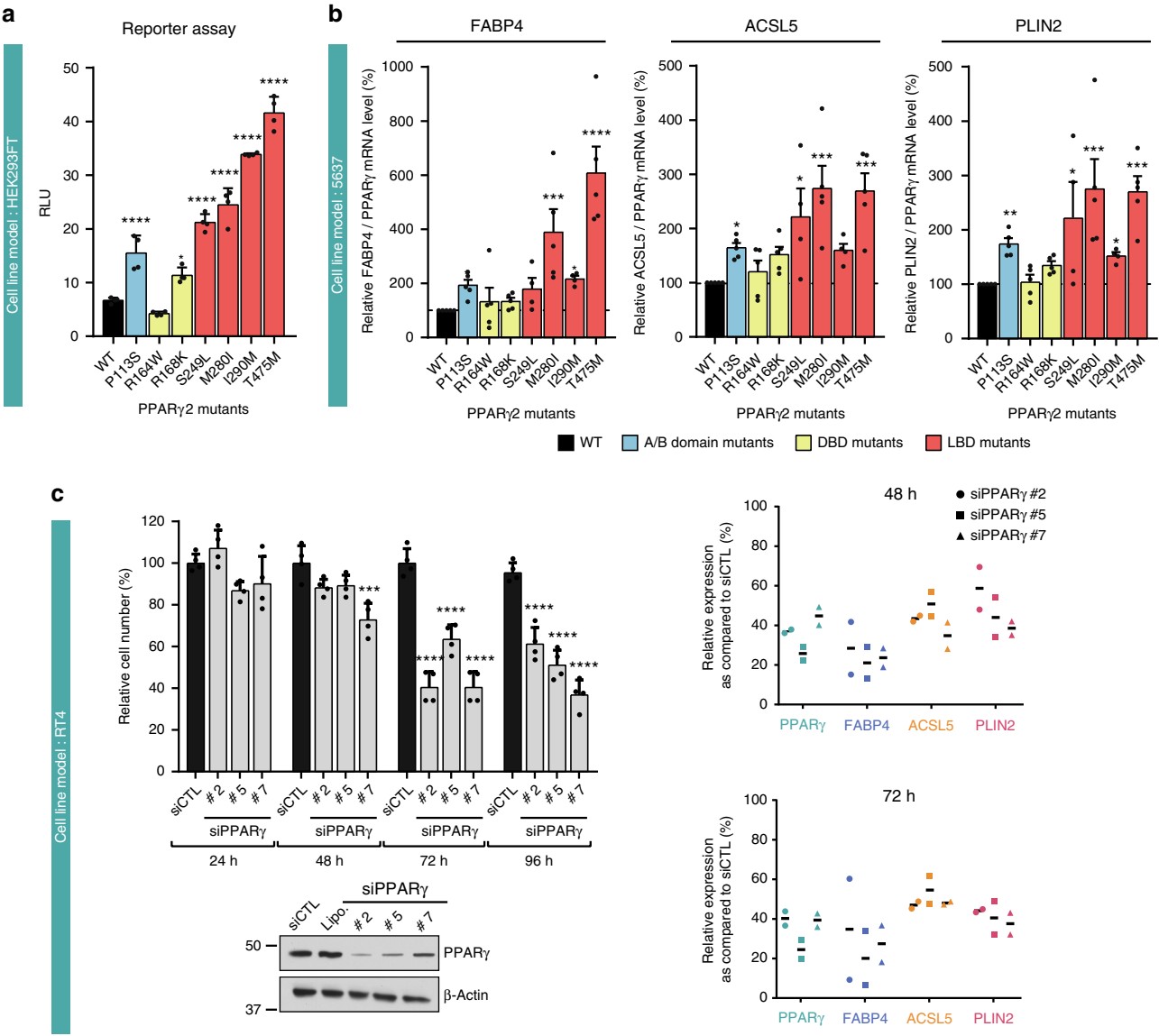

**Fig. 2** Transcriptional activity of recurrent PPARγ mutants. **a** A reporter plasmid containing the firefly luciferase gene under the control of a PPRE-X3-TK promoter was co-expressed in HEK293FT cells with a pcDNA3 vector encoding wild-type (WT) or mutant (P113S, R164W, R168K, S249L, M280I, I290M, T475M) PPARγ2. *Renilla* luciferase, expressed under the control of the CMV promoter, was used to normalize the signal. The data shown are the means ± SD of one representative experiment conducted in sixtuplate. The results for each mutant were compared with those for the WT in Dunnett's multiple comparisons test, *$0.01 < p < 0.05$; ****$p < 0.0001$. **b** 5637 cells were transiently transfected with a pcDNA3 vector encoding WT or mutant (P113S, R164W, R168K, S249L, M280I, I290M, T475M) PPARγ. The expression of all PPARγ forms was checked by western blotting and quantified by RT-qPCR (Supplementary Fig. 1B). The effect of WT PPARγ2 expression on three PPARγ target genes was evaluated by RT-qPCR (Supplementary Fig. 1C). The expression of PPARγ target genes was normalized against PPARγ expression and is expressed as percentage of stimulation relative to the expression induced by WT PPARγ. The data are presented as the mean ± SD of four independent experiments. The results for each mutant were compared with those for the WT in Dunnett's multiple comparisons test: *$0.01 < p < 0.05$; **$0.001 < p < 0.01$; ***$0.0001 < p < 0.001$; ****$p < 0.0001$. **c** PPARγ knockdown with three different siRNAs in RT4 cells harboring the PPARγ T475M mutation. PPARγ expression was evaluated by western blotting (lower left panel) 96 h after transfection, living cells were counted (upper left panel) 24, 48, 72, and 96 h after transfection. Data are presented as means ± SD of three independent experiments performed in duplicate. The results for each mutant were compared with those for the WT in Dunnett's multiple comparisons test: *$0.01 < p < 0.05$; **$0.001 < p < 0.01$; ***$0.0001 < p < 0.001$; ****$p < 0.0001$. The expression levels of three PPARG target genes were assessed by RT-qPCR for two independent experiment at 48 and 96 h after transfection (right panel). Data for each experiment are represented. **a–d** Source data are provided as a Source Data file

effect on PPARγ activity in these two assays. Interestingly, the most frequent mutation, T475M, induced a slightly higher increase in the ligand-independent transcriptional function of PPARγ as compare to any of the recurrent mutations considered (Fig. 2a, b). We used the RT4 cell lines harboring the PPARγ-T475M mutation to demonstrate that, like *PPARG* WT amplification in SD48 and UMUC9 cells[7] and RXRα

-S427F mutation in HT1197[8,13] cells, PPARγ mutations render tumor cell growth PPARγ-dependent and regulate PPARγ target genes expression. Indeed, PPARγ depletion with siRNAs significantly inhibited, in a time-dependent manner, the growth of RT4 cells (Fig. 2c, left panel) and the expression of several known PPARγ target genes (*FABP4*, *ACSL5*, and *PLIN2*) (Fig. 2c, right panel).

**PPARγ LBD mutations favor its interactions with coregulators.** We then used biochemical and biophysical analysis to understand how three mutations located in the ligand-binding domain of PPARγ (M280I, I290M and T475M) promote PPARγ activity. As bacterially expressed PPARγ may contain fatty acids[17], the purified PPARγ LBD WT and mutants (Supplementary Figs. 2 and 3) were analyzed by native electrospray mass spectrometry (Supplementary Fig. 4). The results indicated the absence of any bound ligands. The addition of the potent full PPARγ agonist, GW1929, an N-aryl tyrosine derivative[18] and coactivator peptide led to the formation of ternary complexes for all constructs (Supplementary Fig. 4, right panel). We characterized the interaction between the ligand-binding domains of the PPARγ WT and mutants and the WT RXRα monomer by analytical ultracentrifugation (Supplementary Fig. 5A), which showed that as for the WT, all mutant proteins were able to form heterodimer with

RXRα. To further quantify the interaction between monomeric PPARγ and RXRα ligand-binding domains, we used surface plasmon resonance (SPR) (Fig. 3a and Supplementary Fig. 5B, C). PPARγ T475M, localized at the dimer interface, exhibits an increase by 2-fold of the binding affinity to RXRα compared to PPARγ WT (Fig. 3a), as a consequence of a slower dissociation rate of the dimer (Supplementary Fig. 5B, C). In RXRα the S427F mutation is also spatially localized at the PPARγ/RXRα dimer interface, and it has been shown to similarly stabilize the heterodimer[13]. In contrast, PPARγ M280I shows similar affinity for RXRα as PPARγ WT.

The functional profile of PPARγ, analogous to many other nuclear receptors, is determined by the selective use of transcriptional coregulators that ultimately control the transcriptional output of the target genes. The recruitment of coactivators by PPARγ is primarily determined by the interaction of

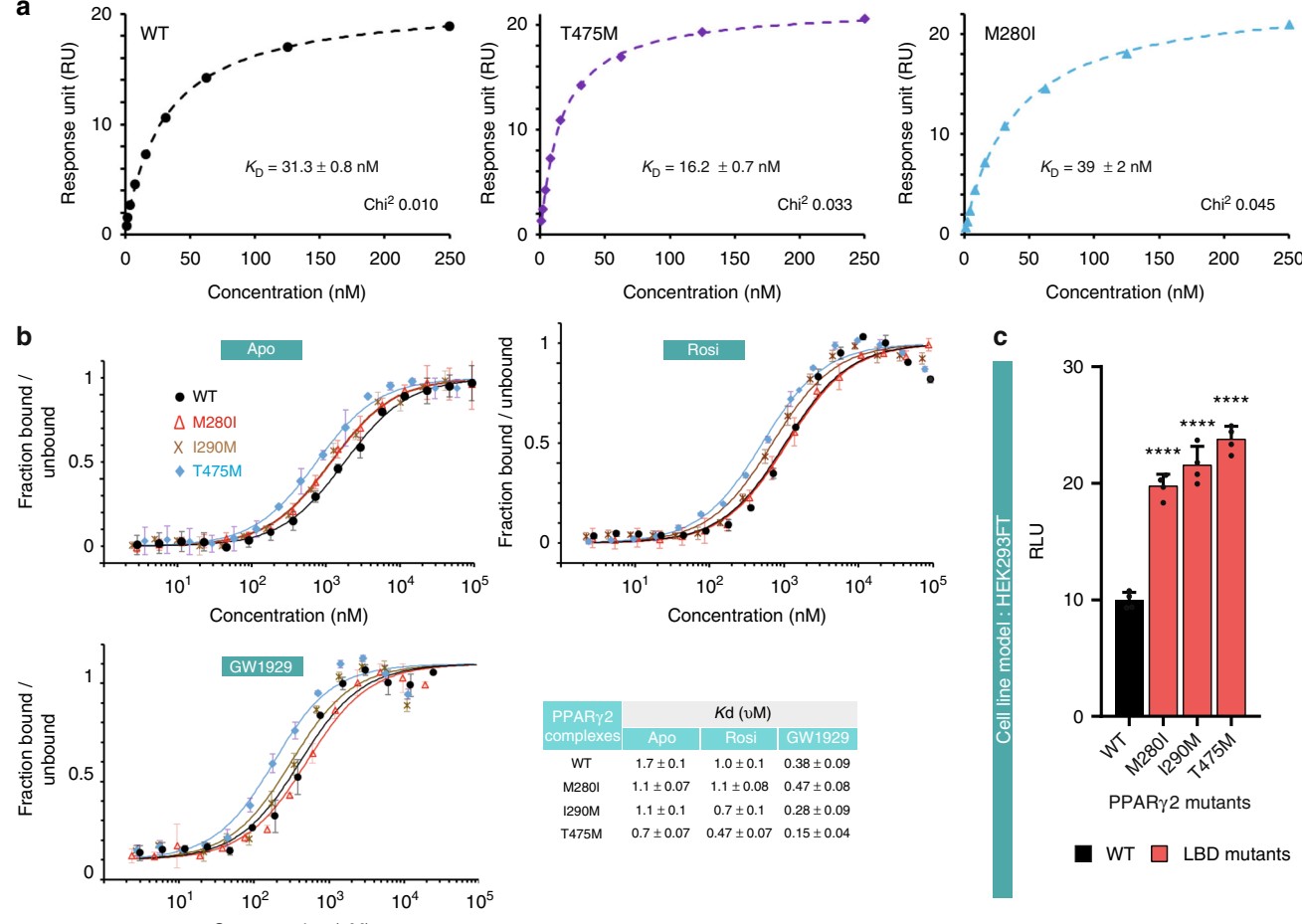

**Fig. 3** Effect of the PPARγ mutations on RXRα and coactivator interactions. **a** Representative data set used for SPR (surface plasmon resonance) analysis of the interactions of PPARγ WT or mutants with RXRα showing enhanced interaction between PPARγ T475M and RXRα. Report point 4 s before injection stop was used for the analysis. Equilibrium responses are plotted as a function of total protein concentration and fit to simple 1:1 binding isotherms. Data analysis by 1:1 kinetic model and mean kinetic parameters and equilibrium dissociation constants are reported in Supplementary Fig. 4. **b** Effect of PPARγ mutations on the PGC1α peptide interaction. The binding affinity of the PGC1α NR1 motif for the purified WT and mutant PPARγ LBDs, as determined by microscale thermophoresis. Unlabeled PPARγ LBD protein was titrated into a fixed concentration of fluorescently labeled PGC1α peptide in the absence of ligand (top left), in the presence of three equivalents of rosiglitazone (top right) or of three equivalents of GW1929 (bottom left). Isotherms averaged over three consecutive measurements and fitted according to the law of mass action to yield the apparent $K_d$. Each plot is representative of at least two independent experiments performed with different batches of protein preparation. **c** Mammalian two-hybrid analysis reveals increased interaction of PPARγ mutants (M280I, I290M and T475M) with MED1 coactivator domain. pG5-Firefly luciferase reporter plasmid was co-expressed with VP16-PPARG (WT or mutants) and with GAL4-DNA-binding domain-fused MED1. *Renilla* luciferase, expressed under the control of the CMV promoter, was used to normalize the signal. The data shown are the means ± SD of one representative experiment conducted in quadruplicate. Results (means ± SD) of three independent experiments are also provided as Supplementary Fig. 16. The results for each mutant were compared with those for the WT in Dunnett's multiple comparisons test, *$0.01 < p < 0.05$; **$0.001 < p < 0.01$. **a**–**c** Source data are provided as a Source Data file

coactivator LXXLL motifs with the receptor LBD. To analyze the functional consequences of the mutations in the LBD, coactivator peptide recruitment by the LBD was monitored. We measured the interaction between monomeric WT or mutant forms of PPARγ and a fluorescently labeled coactivator peptide of PGC1α (PPARGC1A), by MicroScale Thermophoresis. In the absence of ligand, the three mutations enhanced the interaction with coactivator peptide compared to PPARγ WT (Fig. 3b). The addition of a full agonist, rosiglitazone (Fig. 3b), enhanced the interaction between PPARγ and PGC1α coactivator peptide for PPARγ I290M and T475M, but not for PPARγ M280I. The addition of another potent full agonist of PPARγ, GW1929, enhanced yet again the interaction for all mutants, as well as the WT (Fig. 3b). Consistent with the functional data (Fig. 2a, b), monomeric PPARγ T475M had the highest affinity for the coactivator peptides (Fig. 3b). This strongest structural stabilizing effect of T475M mutant on PPARγ-coactivator peptide complex was also observed by ion mobility mass spectrometry which showed significantly improved gas phase stability in collision induced experiments of T475M compared to WT and other PPARγ mutants (Supplementary Figs. 6 and 7). Mammalian two-hybrid assay in HEK293FT cells using VP16-fused PPARγ

(WT, M280I, I290M, and T475M), GAL4-DNA-binding domain-fused coactivator MED1 and pG5-LUC reporter confirmed that in the context of full protein, the three mutations enhanced interaction with MED1 coactivator domain compared to PPARγ WT (Fig. 3c). Together, these data suggest that the three mutations considered, M280I, I290M and T475M, promote the adoption of an agonist conformation by PPARγ in the absence of ligand, thereby enhancing coactivator interaction.

**PPARγ LBD mutations stabilize an active conformation**. To gain structural insight into the mechanism responsible for the increases in activity and coactivator interaction, we analyzed the structures of the PPARγ LBD T475M, M280I and I290M mutants (Supplementary Table 5). The PPARγ T475M LBD mutant was crystallized in its apo form and in complex with GW1929 and the PGC1α coactivator peptide (Fig. 4a and Supplementary Fig. 8). In both functional states, PPARγ T475M crystallized as a homodimer. More than 150 crystal structures of the PPARγ LBD have been deposited in the Protein Data Bank (www.rcsb.org)[19], and in many of them, the PPARγ LBD crystallized as a homodimer displaying the canonical dimer interface observed in the hetero-dimer complex. Of these structures, the apo protein and some

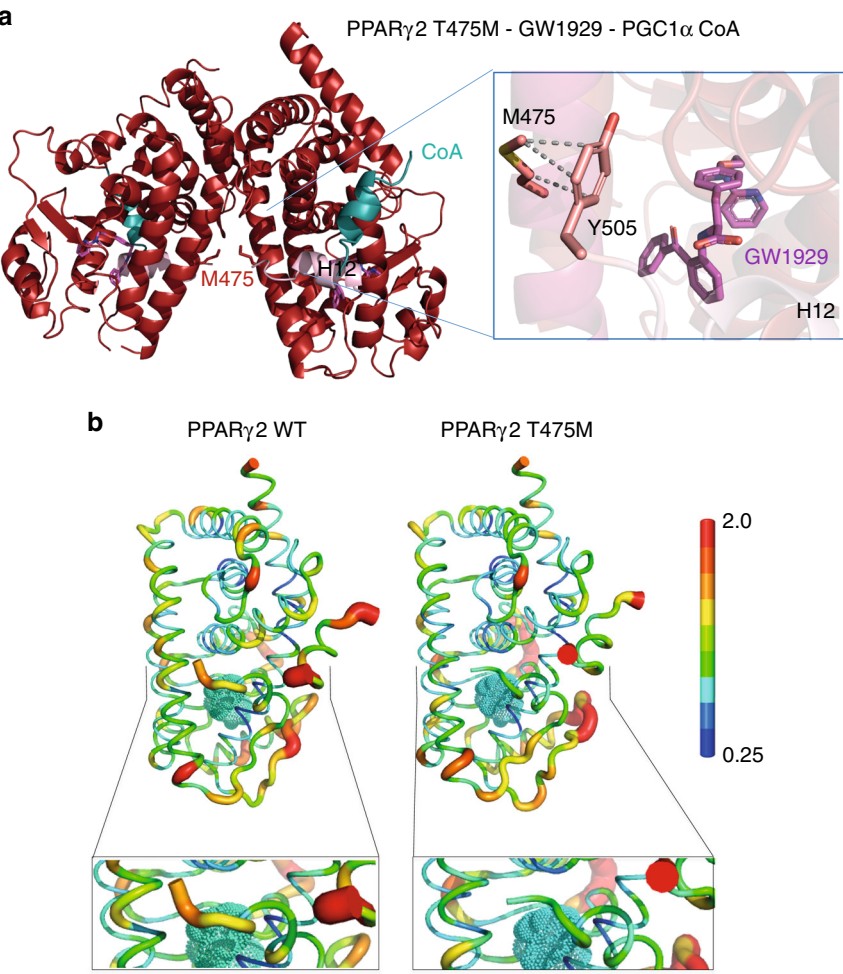

**Fig. 4** Structural properties of PPARγ T475M. **a** Crystal structure of the PPARγ T475M LBD (red) in complex with GW1929 (violet) and the PGC1α coactivator peptide (blue). The C-terminal H12 helices are shown in light pink. Overall fold of the homodimer complex, showing the T475M mutations at the dimer interface as a stick representation. Right: Close-up of the regions around the mutation, showing its interactions with the terminal Y505 residue responsible for stabilizing the agonist conformation. **b** The PPARγ T475M mutation modulates the structural dynamics of the LDB. Mean structure and atomic fluctuations of the holo PPARγ WT LBD-coactivator (left) and holo PPARγ T475M LBD-coactivator (right) complexes, with the rosiglitazone ligand (in cyan), and the scale of flexibility shown. Bottom: Close-up of the C-terminal H12 helix of the WT and mutant LBDs, as determined from molecular dynamics simulations

complexes with partial agonists display one monomer in an active conformation and one monomer in an inactive conformation with a different positioning of helix 12[20,21]. Helix 12 is key regulatory structural element in the activation function 2 interacting with coregulators. These 2 conformations are generally designated as fully active and inactive[21]. Although these helix 12 conformations observed in the crystal structures are influenced to some degree by crystal packing, they reflect the dynamic character of helix 12 in absence or presence of a partial agonist ligand. This was further confirmed by solution structural methods[22]. Interestingly, in contrast to the structures of WT PPARγ in its apo form, PPARγ T475M apo displays an agonist conformation in both homodimer LBDs, indicating a stabilization of helix 12 in an active conformation even in absence of any agonist ligand (Supplementary Fig. 8).

In a similar way, the holo PPARγ T475M-GW19129-PGC1α ternary complex crystallized as a homodimer (Fig. 4a) and the structure agrees with the crystal structure of another N-aryl tyrosine derivative complex, GI262570[23]. The GW1929 ligand binds in a U-shaped conformation with the carboxyl group of GW19219 making hydrogen bonds with S317, H351, H477, and Y501 (Supplementary Fig. 9). The pyridinyl tail is directed towards the β-sheet and its nitrogen atom makes an H-bond with a bound water molecule. The benzophenone attached to the N-aryl tyrosine, also present in GI262570, forms additional hydrophobic interactions, not available to rosiglitazone, explaining their increased PPARγ binding affinity and enhanced coactivator interaction[18,22,24]. The mutated residue T475M interacts with the C-terminal Y505 (Fig. 4a) stabilizing helix 12 into the active conformation and leading to a more stable interaction with the coactivator peptide. Interestingly, a similar interaction stabilizing PPARγ in the active agonist conformation was observed in the crystal structure of a heterodimer complex of holo WT PPARγ and the RXRα S427F mutant[13]. A similar observation was made by molecular dynamics (MD) simulations of the WT[20] and T475M PPARγ LBD (Fig. 4b). Starting from a previously determined crystal structure of the WT protein[20], the point mutation was modeled into the WT structure. Separate 100 ns simulations of monomeric WT and T475M mutant PPARγ LBDs in both apo form and in complex with rosiglitazone were performed. We found that the T475M mutation decreased the structural flexibility of the protein, particularly that of the C-terminal H12 helix (Fig. 4b), through direct interaction between the M475 and the C-terminal Y505, as observed in the structure of PPARγ T475M in complex with GW1929. In the WT structure, the analogous interaction (between T475 and Y505) does not appear; the M475–Y505 interaction formed during the molecular dynamics simulations (Supplementary Fig. 10). The decrease in the flexibility of the C-terminal H12 helix in the T475M mutant led to a lower flexibility of the coactivator peptide, resulting in a more stable complex. Free energies of coactivator interaction were calculated by the MM/GBSA[25] method on 100 structures taken from the MD simulations at regular intervals. This analysis yielded binding energy estimates of −43 ± 7 kcal/mol and −48 ± 5 kcal/mol for WT PPARγ and for the T475M mutant, respectively, consistent with the trends observed in the experimental data of coactivator interaction. In this application, the MM/GBSA binding energies are used to rank interactions and not to calculate absolute binding-free energies.

As the T475M mutation is located at the dimer interface, we modeled the structure of the heterodimer formed by PPARγ T475M (Supplementary Fig. 11). The results suggest that T475M will strengthen the heterodimer through interactions with L430 and S427 in contrast to WT, in agreement with the experimental data. Molecular dynamics simulations of a PPARγ-T475M/RXRα heterodimer model showed that the heterodimer is more structurally stable than the WT and, as observed in the monomer simulations, the coactivator peptide displayed reduced flexibility (Supplementary Fig. 11C). Thus, our structural analysis of PPARγ T475M indicates that the mutation stabilizes PPARγ helix 12 in the active conformation as well as the heterodimer with RXRα, leading to a stronger binding to coactivators. This mechanism may explain PPARγ-dependent transcription program activation.

The two other mutants of interest here are localized in the Ω loop of the PPARγ LBD; M280I is in the helix 2′ of the Ω loop and I290M is in the loop between helix 2′ and helix 3. The Ω loop is often poorly defined in the electron density maps of the published structures of PPARγ, adopting varying conformations depending on the nature of the ligand. It is thought to serve as a gate to the ligand-binding pocket[26] and also serve as an alternative ligand-binding site for some ligands[27,28]. The PPARγ M280I LBD crystallized in complex with GW1929 and PGC1α peptide as a homodimer (Fig. 5a) and PPARγ I290M in complex with GW1929 as a monomer (Supplementary Fig. 12B). The conformation and interactions of GW1929 in the PPARγ LBPs is similar in all complexes (Supplementary Fig. 9A). While helix 2′ is similarly positioned in all complexes, the flexible loop connecting helix 2′ and helix 3 shows different conformations and is partially resolved in some complexes. The M280I mutant stabilizes helix 3, notably through specific interactions with V305 (Fig. 5a), that, in turn stabilizes PPARγ in the agonist conformation. Recall that the coactivator interaction is via the activation function 2 coregulator surface composed of helix 3, 4, 5, and helix 12[20]. For the I290M mutant, the surrounding loop connecting helix 2′ and helix 3 shows some flexibility as indicated by the poor density of some side chains residues and higher temperature factors. However, the structural analysis of I290M–GW1929 complex indicates that I290M interacts more strongly with I369, which contacts the ligand (Supplementary Fig. 12B) and may stabilize the β-sheet, as well as helix 3. Following the protocol used in the previously described simulations, molecular dynamics simulations of the M280I and I290M LBDs were carried out. From the average structure calculated over the last 10 ns of each simulation, we found interactions that stabilized the Ω loop region via helix 3 and further stabilization of H12 activation function 2 for both the M280I and the I290M mutants. In the case of M280I mutant, I280 formed a stable interaction with V305 of H3, the N-terminal end of H3. This interaction was also seen in the crystal structure of the M280I mutant described above. In the I290M variant, M290 forms an interaction with Phe315 of H3, which interacts with His494 at the N-terminal end of H12. Stabilization of H12 leads to a stabilization of the coactivator peptide, as indicated by the decreased atomic fluctuations shown in Fig. 5b, which shows the rms fluctuations mapped onto the 3D surface of the proteins complexes. The free energies of coactivator binding to M280I and I290M were also estimated by the MM/GBSA method on structures taken from the MD simulations at regular intervals. The binding energies, −40 ± 7 kcal/mol and −45 ± 7 kcal/mol for M280I and I290M, respectively, are consistent with the trends observed in the experimental data of coactivator interaction. In particular, the mutant I290M shows an enhanced binding free energy with respect to the WT.

We used hydrogen/deuterium exchange coupled to mass spectrometry (HDX–MS) to investigate complementary structural information about the effect of T475M and M280I mutations on PPARγ structure. Of note, due to lack of reproducibility in HDX–MS data acquisition on I290M, these data were excluded from the present study. HDX–MS is a powerful readout to monitor protein conformational dynamics at peptide resolution, by monitoring the D exchange of amide hydrogens with deuterated solvent[24]. In order to evidence PPARγ point mutation

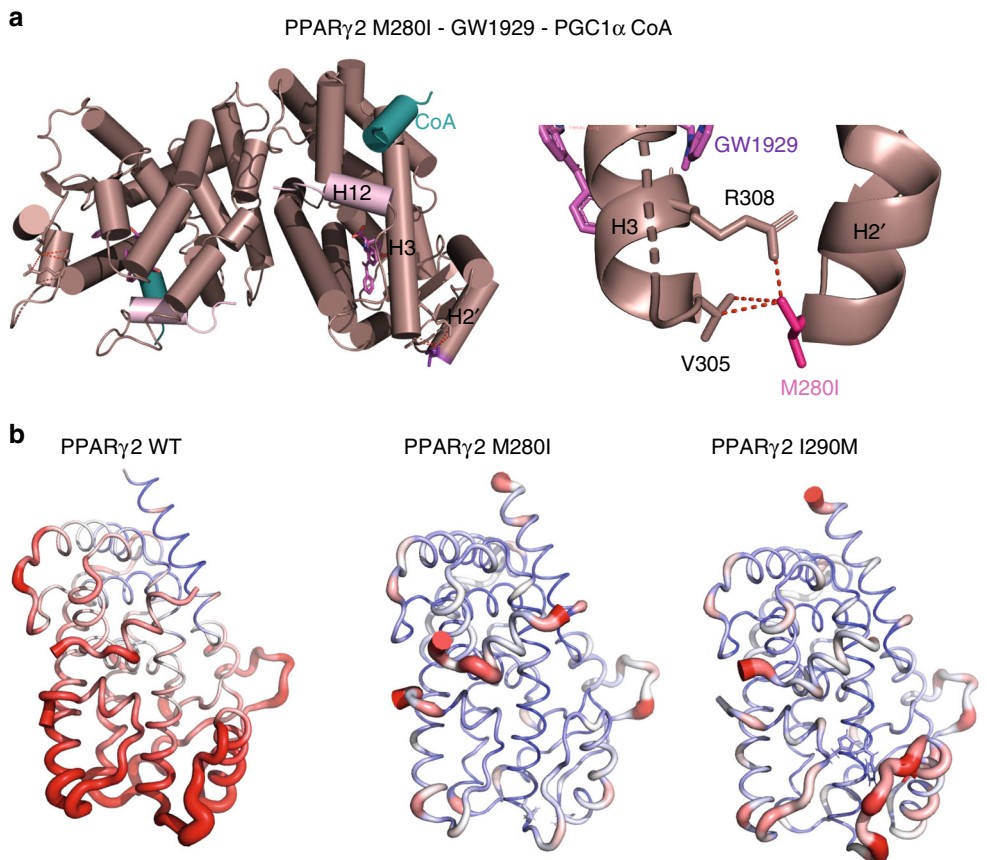

**Fig. 5** Impact of PPARγ M280I and I290M mutations on the structure and dynamics. **a** Crystal structure of the PPARγ M280I LBD (plum) in complex with GW1929 (light red) and the PGC1α coactivator peptide (blue). The C-terminal H12 are shown in light pink. Right: Close-up of the regions around the mutation, showing its interactions stabilizing helix 3. **b** The atomic fluctuations calculated from the molecular dynamics simulation (last 10 ns) of the holo form of PPARγ complexes with rosiglitazone and coactivator peptide. The fluctuations are mapped onto the 3D surface of the proteins complexes. The color scale goes from blue (less flexible) to red (most flexible) between 0 and 2.5 Å. The thickness of the tube also reflects the flexibility (thicker corresponds to more flexible). Left: WT; middle: M280I; right: I290M

effects on secondary structures of PPARγ/PGC1α complexes, we performed pairwise comparison of deuterium uptakes of PPARγ WT/PGC1α to PPARγ mutant/PGC1α (Supplementary Fig. 13). For this, relative fractional uptakes (RFU) of PPARγ mutants/PGC1α were subtracted from the PPARγ WT/PGC1αα RFU and the significance of the RFU difference plots was validated using the MEMHDX software[29] ($p < 0.01$, Supplementary Fig. 13). Overall, PPARγ mutant proteins presented a lower level of deuterium uptake compared to WT when bound to PGC1α, highlighting the fact that PPARγ mutant/PGC1α complexes present several regions less flexible or less exposed to solvent than PPARγ WT/PGC1α complex (Supplementary Fig. 14).

Comparison of PPARγ T475M/PGC1α to PPARγ WT/PGC1α RFUs revealed that different regions of the T475M mutant presented a significant lower deuterium uptake (Supplementary Fig. 14, upper panel), including H2, H2′, N-terminal H3, H6, N-terminal H8, helix H10, and the H12 helix. These regions are thus more protected from deuterium exchange for PPARγ T475M compared to PPARγ WT, which might be in favor of a decreased conformational flexibility of PPARγ T475M compared to PPARγ WT, in agreement with MD simulation data.

For PPARγ M280I a significantly different behavior was observed compared to PPARγ T475M (Supplementary Fig. 14, middle panel). Decreased deuterium uptake was observed around the mutation site (from the C-terminal H1 towards the mutation site and until H3, including the Ω loop). In addition, the β-sheet region, H6 helix, the loop between H8 and H9 and finally the

region encompassing H10, H11, and H12 helices showed also significant decrease in deuterium uptake, which might correspond to a decreased conformational flexibility. Our HDX–MS data for the PPARγ M280I are consistent with MD simulations high-lighting a stabilization of the Ω loop, H3 and H12 helices in the M280I mutant protein, compared to WT. Altogether, HDX–MS data are thus in agreement with our structural analysis of the structural elements involved in agonist conformation stabilization of the T475M and M280I mutant proteins.

## Discussion

Taken together, our study highlighted the existence of recurrent driver gain-of-function PPARγ mutations in a particular type of cancer: luminal bladder tumors. Frequent PAX8-PPARγ fusions have already been reported in follicular thyroid tumors (30% of these tumors), but these proteins, which also have a proto-oncogenic role, are likely to act in a dominant negative manner[30]. The P113S mutation enhanced PPARγ activity, probably by inhibiting S112 phosphorylation by MAP kinases, as shown for the only gain-of-function mutation of PPARγ reported to date, P113G, which is associated with obesity[31]. We have also identified two recurrent mutations affecting the DNA-binding domain of PPARγ, but the functional impact of these mutations was not as clear as for the other mutants and requires further investigation. However, we clearly showed that the three recurrent mutations affecting the ligand-binding domain of PPARγ altered the

conformation and structural dynamics of the protein, stabilizing helix H12 and so promoting the adoption of the active form by the receptor even in the absence of ligand. These mutations also favor the recruitment of coactivators. Such effects have already been described for mutations affecting the ligand-binding domain of another hormone receptor, ER, in breast cancer, these mutations however being found exclusively in hormone-resistant metastases, not in primary tumors[32]. Mutations affecting the ligand-binding domain of the androgen receptor have also been observed in metastatic or castration-resistant prostate cancers and these mutations enable the protein to bind other ligands[33]. Our demonstration that the activation of PPARγ by point mutation, like PPARG amplification[7] or RXRα mutation[8,13], confers bladder cancer cells a PPARγ dependency, provides additional genetic evidence for a pro-tumorigenic role of PPARγ in bladder cancer and strengthens the importance of the PPARγ/RXRα pathway in luminal bladder cancer. RXRα hotspot mutations appear to be specific to bladder cancer. By contrast, the PPARγ gain-of-function mutations identified here are also observed in other types of cancer. In particular, T475M mutations are observed in four different types of cancer (Supplementary Table 4). Moreover, other recurrent mutations not identified in bladder cancer have been observed in different types of cancer, including melanoma and prostate carcinoma (Supplementary Table 6), and these mutations may also likely be activating mutations. The activation of PPARγ by mutations may, therefore, result in PPARγ-dependence not only in bladder cancer, but also in other types of cancer, reinforcing the importance of developing new pharmacological approaches targeting the ligand-independent activity of PPARγ in tumors. Such treatments would also favor the immune response in bladder tumors[13]. A better understanding of the molecular basis of the pro-oncogenic activity of PPARγ in bladder luminal tumors might also make it possible to propose alternative therapeutic options for targeting this pathway without directly inhibiting PPARγ, which could lead to diabetes or lipodystrophy[34].

## Methods

**Human bladder samples**. We used DNA extracted from 359 bladder tumors. The flash-frozen tumor samples were stored at −80 °C immediately after transurethral resection or cystectomy. All tumor samples contained more than 80% tumor cells, as assessed by the hematoxylin and eosin (H&E) staining of histological sections adjacent to the samples used for DNA extraction. All subjects provided informed consent and the study was approved by the institutional review boards of the Henri Mondor, Foch, Institut Gustave Roussy and Strasbourg Hospitals.

**Sanger sequencing**. The coding exons and splice junctions of PPARG and RXRA were amplified from genomic DNA by PCR with gene-specific primers available on request and sequenced by the Sanger method. For tumors, the somatic status of the identified mutation was confirmed by sequencing normal DNA from blood. Sequencing was performed for 25 bladder cell lines: 5637, BFTC-905, CAL29, EJ138, HT1376, J82, JMSU1, KK47, L1207, MGHU3, RT112, RT4, SCaBER, SD48, SW1710, T24, TCCSup, UMUC1, UMUC5, UMUC6, UMUC9, UMUC10, UMUC16, VMCUB1, and VMCUB3. The identity of the cell lines used was checked by analyzing genomic alterations with comparative genomic hybridization arrays (CGH array), and FGFR3 and TP53 mutations were checked with the SNaPshot technique (for FGFR3) or by classical sequencing (for TP53). The results obtained were compared with the initial description of the cells. We routinely checked for mycoplasma contamination.

**Calculation of a PPARγ activation score**. Based on our previously described PPARγ activation signature encompassing 148 genes[7], we defined a subset of 77 genes that were also significantly more expressed in the 25% of tumors with the strongest PPARγ expression than in the 25% of tumors with the lowest levels of PPARγ expression in TCGA datasets ($N = 405$ tumors)[10]. The PPARγ activation score is the mean of the centered expressions of these 77 genes for each tumor.

**Materials and chemicals**. Rosiglitazone and GW1929 were purchased from Tocris Bioscience. The fluorescent PGC1α peptide (137-EAEEPSLLKKLLLAPA-152) was purchased from Thermo-Fisher. The PGC1α peptide (139-EEPSLLKKLLLAPA-152) was synthesized by Pascal Eberling (IGBMC peptide synthesis common facility).

**Plasmid constructs**. The pcDNA3-PPARγ2 and PPRE-X3-TK-luc were generously provided by Pr. Chatterjee (Institute of Metabolic Science, IMS, Cambridge) and Bruce Spiegelman (Addgene plasmid #1015), respectively. We used pcDNA3.1-PPARγ2 and the QuikChange II Site-Directed Mutagenesis Kit (Agilent Technologies) according to the manufacturer's protocol, to generate all the mutations. Mutations were confirmed by DNA sequencing. The GAL4-DNA-binding domain cloning vector pM and the activation-domain cloning vector pVP16 are part of the Mammalian Matchmaker Two-Hybrid Assay kit (BD Biosciences Clontech). The construct pM-MED1 (510–787) expressing the Gal4 DBD-MED1 nuclear receptor interacting domain was provided by Lieve Verlinden (KU Leuven, Belgium)[35].

**Cell culture and transfection**. The HEK293FT human cell line and the RT4 and 5637 human bladder tumor-derived cell lines were obtained from DSMZ (Heidelberg, Germany). HEK293FT cells were cultured in DMEM, whereas 5637 and RT4 cells were cultured in RPMI. Media were supplemented with 10% fetal calf serum (FCS). Cells were incubated at 37 °C, under an atmosphere containing 5% $CO_2$.

For reporter gene assays, HEK293FT cells were plated in 96-well plates (30,000 cells per well) and transfected with 30 ng pcDNA3-PPARγ2 (WT or mutated), 50 ng PPRE-X3-TK-luc and 6 ng pRL-SV40 (Promega), in the presence of the Fugene HD transfection reagent (Promega), in accordance with the manufacturer's protocol. Luciferase activity was determined 48 h later, with the Dual-Glo® Luciferase Assay System (Promega), according to the manufacturer's instructions, and the results obtained were normalized with the Renilla luciferase signal obtained with the pRL-SV40 plasmid.

For PPARγ2 overexpression in the 5637 cell line, we used six-well plates, 250,000 cells seeded per well. These cells were transfected 24 h later with 2.5 μg of pcDNA3-PPARγ2 (WT or mutated) in the presence of the Fugene HD transfection reagent (Promega). RNAs were extracted with the RNA easy mini kit (Qiagen) and proteins were extracted by cell lysis in Laemmli buffer (50 mM Tris–HCl (pH 7.5), 250 mM NaCl, 1% SDS) supplemented with protease inhibitors and phosphatase inhibitors (Roche) 48 h after transfection.

For transfection with siRNA, RT4 cells were used to seed six-well plates at a density of 300,000 cells/well. Cells were transfected with 20 nM siRNA in the presence of Lipofectamine RNAi Max reagent (Invitrogen), in accordance with the manufacturer's protocol. siRNAs were purchased from Qiagen. For the control siRNA, we used a control targeting luciferase (SI03650353). We used three PPARγ siRNAs designed to knockdown the expression of all known mRNA isoforms. The sense-strand sequences were: PPARG siRNA#2: GACAAAUCACCAUUCGUUATT, PPARG siRNA#5: GCGACUUGGCAAUAUUUAUTT and PPARG siRNA#7: CGGAGAACAAUCAGAUUGATT.

Cells were detached from the plates with trypsin 96 h after transfection, counted using a hemocytometer and lysed in Laemmli buffer (50 mM Tris–HCl (pH 7.5), 250 mM NaCl, 1% SDS) supplemented with protease inhibitors and phosphatase inhibitors (Roche).

For mammalian two-hybrid assay, HEK293FT cells were plated in 96-well plate (30,000 cells per well) and transfected with 20 ng pV16-PPARγ2 (WT or mutated), 20 ng pM-MED1, 50 ng pG5-luc (Promega) reporter plasmid and 6 ng pRL-SV40 (Promega), in the presence of the Fugene HD transfection reagent (Promega), in accordance with the manufacturer's protocol. Luciferase activity was determined 48 h later, with the Dual-Glo® Luciferase Assay System (Promega), according to the manufacturer's instructions, and the results obtained were normalized with the Renilla luciferase signal obtained with the pRL-SV40 plasmid.

**Immunoblotting**. The 5637 and RT4 cell lysates were clarified by centrifugation. The protein concentration of the supernatants was determined with the BCA protein assay (Thermo Scientific). 10 μg of proteins were resolved by SDS-PAGE in a 4–15% polyacrylamide gels, electrotransferred onto Biorad nitrocellulose membranes and analyzed by incubation with primary antibodies against PPARγ (Abcam #ab41928, used at 1/1000) and β-actin (Sigma Aldrich #A2228, used at 1/25,000). Horseradish peroxidase-conjugated anti-mouse IgG (Cell Signaling Technology # 7074, used at 1/3000) was used as the secondary antibody. Protein loading was checked by staining the membrane with Amido Black after electroblotting. Uncropped scans of the western blot are supplied Supplementary Fig. 15.

**Real-time reverse transcription-quantitative PCR**. Reverse transcription was performed with 1 μg of total RNA, and a high-capacity cDNA reverse transcription kit (Applied Biosystems). cDNAs were amplified by PCR in a Roche real-time thermal cycler, with the Roche Taqman master mix (Roche) and Taqman probe/primer pairs as follows:

PLIN2:
Forward primer—TCTGAATCAGCCATCAACTCAG;
Reverse primer—GTGCTGGCCACAGAATCC;
Roche Taqman probe no. 57
ACSL5:
Forward primer—TGTCCACTTCAGTCATGACATTCT;
Reverse primer—TCCAGTCCCCAGGTAATGTAA;
Roche Taqman probe no. 83

FABP4:
Forward primer—GGATGATAAACTGGTGGTGGA;
Reverse primer—CACAGAATGTTGTAGAGTTCAATGC
Roche Taqman probe no. 85
PPARγ and TBP were amplified by PCR in a Roche real-time thermal cycler with Roche SYBR green master mix and primers, as follows:
PPARγ:
Forward primer—GCCCAAGTTTGAGTTTGCTG;
Reverse primer—TCAATGGGCTTCACATTCAGC;
TBP:
Forward primer—TTGCTGCGGTAATCATGAGG;
Reverse primer—TTTTCTTGCTGCCAGTCTGG

We used the *4326322E* assays on demand for TBP (encompassing primers and *Taq*man probes) purchased from Applied Life Technologies.

Relative gene expression was analyzed by the delta delta Ct method, with TBP as the reference.

**Biochemistry.** The sequences encoding the ligand-binding domains of the His-hPPARγ (231–505) and His-mRXRα (228–467) receptors were inserted into pET15b. Point mutations were introduced into *PPARγ* with the QuikChange II XL Site-Directed Mutagenesis kit (Agilent), in accordance with the manufacturer's instructions.

The corresponding proteins were produced in *Escherichia coli* BL21 DE3 by overnight incubation at 22 °C after induction with 1 mM IPTG at an $OD_{600}$ of ~0.8. Soluble proteins were purified by Ni-NTA chromatography followed by size exclusion chromatography on a Superdex 200 (GE) column equilibrated in 25 mM Tris–HCl, pH 8.0, 200 mM NaCl, 5% glycerol, and 1 mM TCEP. The proteins were concentrated to 3–6 mg/ml with an Amicon Ultra 10 kD MWCO. Purity and homogeneity of all proteins were assessed by SDS and Native Page and for PPAR proteins by denaturing and native electrospray ionization mass spectrometry (Supplementary Figs. 3 and 4).

**Crystallization, data collection, and structure refinement.** Crystallization trials were performed with either apo protein or with the protein in complex with a threefold excess of GW1929 or a threefold excess of GW1929 and PGC1α peptide. The crystallization experiments were performed by sitting drop vapor diffusion at 290 K, mixing equal volumes (200 nl) of protein at 5 mg/ml and reservoir solution. For all crystal structures, the data were indexed and integrated with XDS[36] and scaled with AIMLESS[37,38]. The structure was solved by molecular replacement in PHASER[39] and refined with PHENIX[40] and BUSTER[41] with TLS refinement, followed by iterative model building in COOT[42].

Crystals of PPARγ M280I-GW1929-PGC1α were grown in 20% PEG 550 MME, 10% PEG 20000, 30 mM sodium fluoride, 30 mM sodium bromide, 30 mM sodium iodide, 0.1 M sodium Hepes/MOPS pH 7.5, transferred to artificial mother liquor containing 15% glycerol and flash-cooled in liquid nitrogen. X-ray diffraction data were collected at PX1 beamline of the SOLEIL synchrotron with a wavelength of 0.979 Å. The final structure was refined to $R_{work}$ and $R_{free}$ values of 17.5% and 22.6%, respectively, with excellent geometry (97.99% of residues in favored region of the Ramachandran plot, 2.01% in the allowed region, and 0.0% outliers).

Crystals of PPARγ T475M-GW1929-PGC1α were grown in 25% PEG 3350, 0.2 M ammonium sulfate, 0.1 M Bis–Tris pH 6.5, transferred to artificial mother liquor containing 15% glycerol and flash-cooled in liquid nitrogen. X-ray diffraction data were collected at the ID23-1 beamline of ESRF with a wavelength of 0.9724 Å. The final structure was refined to $R_{work}$ and $R_{free}$ values of 15.9% and 19.88%, respectively, with excellent geometry (98.23% of residues in favored region of the Ramachandran plot, 1.77% in the allowed region, and 0.0% outliers).

Crystals of PPARγ T475M apo were grown in 0.5 M ammonium sulfate, 0.1 M PIPES pH 7, 0.9 M disodium tartrate. Crystals were transferred to artificial mother liquor containing 15% glycerol and flash-cooled in liquid nitrogen. X-ray diffraction data were collected at the ID23-1 beamline of ESRF with a wavelength of 0.9724 Å. The final structure was refined to $R_{work}$ and $R_{free}$ values of 19.5% and 23.48%, respectively, with good geometry (96.05% of residues in favored region of the Ramachandran plot, 3.77% in the allowed region, and 0.19% outliers).

Crystals of PPARγ WT-GW1929-PGC1α were grown in 20% PEG 5000 MME, 0.1 M Bis–Tris pH 6.5. Crystals were transferred to artificial mother liquor containing 15% glycerol and flash-cooled in liquid nitrogen. X-ray diffraction data were collected at the ID30A beamline of ESRF with a wavelength of 0.968 Å. The final structure was refined to $R_{work}$ and $R_{free}$ values of 21.9% and 25.2%, respectively, with excellent geometry (96.53% of residues in favored region of the Ramachandran plot, 3.47% in the allowed region, and 0.00% outliers).

Crystals of PPARγ I290M–GW1929 complex were grown in 25% PEG 3350, 0.2 M NaCl, 0.1 M Hepes pH 7.5 and were transferred to artificial mother liquor containing 35% PEG 3350 before flash-cooling. X-ray diffraction data were collected at the ID23-1 beamline of ESRF with a wavelength of 0.9724 Å. The final structure was refined to $R_{work}$ and $R_{free}$ values of 16.52% and 20.89%, respectively, with excellent geometry (98.54% of residues in favored region of the Ramachandran plot, 1.46% in the allowed region, and 0.00% outliers).

Data collection and refinement statistics are provided in Supplementary Table 5. GW1929 and side chains of the mutated residues could be modeled

with confidence in all chains and all complexes as shown into the Polder omit maps[43] displaying reduced model bias and exclusion of solvent molecules (Supplementary Figs. 9A and 12A). All structural figures were prepared with PyMOL (www.pymol.org/).

**Molecular dynamics simulations.** hPPARγ PDB:2PRG[20] was used for all calculations. For the WT holo form, we used chain A, the ligand and coactivator peptide segment from the crystal structure. For the WT apo form, we used the same structure, but we removed the ligand and the coactivator peptide from the structure. For the corresponding mutant forms, the point mutation was built into this experimental crystal structure with CHARMM program version c37b1[44]. All forms were subjected to the PROPKA program[45] to determine the protonation states of titratable residues. These calculations indicated that the H217, H266, H323, His449, and H466 residues of chain A were protonated on the Nε, but that H425 could be considered doubly protonated. We modified the PDB file accordingly to incorporate the correct protonation states. Hydrogen atom positions were positioned with the HBUILD[46] module of the CHARMM program. The parameters for rosiglitazone were determined in a previous study[47], MD simulations were performed with NAMD software[48] and the CHARMM all-atom force field, version 36[49]. Crystal water molecules associated with the hPPARγ protein were retained. Energy minimization was performed for the protein, in 700 steps, with the steepest descent (SD) algorithm in CHARMM software. Non-bonded interactions were truncated, using a 14 Å cutoff distance, using switch and shift functions for van der Waals and electrostatic forces, respectively. The protein was then placed in a cubic TIP3P explicit water box of dimensions $80 \times 80 \times 80$ Å. Chloride and sodium counter ions were added to the water box to neutralize the charge of the system and additional ions were added to yield a final concentration of 0.15 M. MD simulations were initiated with two rounds of energy minimization and heating while the atomic coordinates of the protein complex were fixed. In the first phase, water was minimized by 1000 steps of conjugate gradient (CG) and then heated to 600 K. In the second phase, the water was minimized by 250 steps of CG, followed by heating to 300 K. The constraints on the protein were then removed and the entire system was subjected to energy minimization through 2000 steps of CG and subsequent heating to 300 K. The system was then equilibrated for 150 ps. With a 1 fs time step, the production phase extended over a total of 107 ns. This protocol was used for all forms studied. The root-mean-square deviation (RMSD) and root-mean-square fluctuation (RMSfl) were calculated using the CHARMM program and the coactivator peptide binding energies were estimated by an MM/GBSA procedure[25] using the CHARMM program combined with in-house programs.

**Microscale thermophoresis.** Measurements were performed with a Monolith NT.115 instrument (NanoTemper Technologies GmbH, Munchen, Germany). The PPARγ complexes were prepared in 20 mM Tris pH 8.0, 200 mM NaCl, 1 mM TCEP, 0.05% Tween 20. Each measurement consists of 16 reaction mixtures where the fluorescent-labeled peptide concentration was constant (150 nM) and serial dilutions of PPARγ LBD from a concentration of 100 μM down to 2 nM. Measurements were made with standard glass capillaries (Nanotemper) at 25 °C, at 30% LED excitation and 40–80% MST power, with a laser-on time of 30 s and a laser-off time of 5 s. NanoTemper Analysis 2.2.4 software was used to fit the data and to determine the $K_d$.

**Analytical ultracentrifugation.** Sedimentation velocity experiments were performed at 4 °C, with a Beckman Optima XL-A analytical ultracentrifuge equipped with absorbance optics and an An50-Ti rotor in a buffer containing 20 mM Tris, pH 8.0, 300 mM NaCl, 5 mM Glycerol, 1 mM TCEP. For this analysis, the heterodimer was formed by mixing the monomeric PPARγ and RXRα proteins into a 1:1 molar ratio. Protein concentrations were in the range of 0.8 to 0.9 mg/ml. The sedimentation velocity analysis was conducted at 165,500×g. Values were normalized to standard conditions by correcting for buffer density and viscosity. Sedimentation coefficient distributions were calculated from the sedimentation velocity data using the SEDFIT software program (www.analyticalultracentrifugation.com).

**Surface plasmon resonance.** The SPR measurements were performed by Biacore T100 sensitivity enhanced T200 equipment (GE Healthcare) using CM5 series S sensor chip. The RXRα LBD monomer was immobilized on the chip surface using standard amino-coupling protocol in 10 mM Na-acetate buffer pH 5.5. The resulting immobilized RXR was in the range of 400–500 response units. The running buffer was 50 mM Tris pH 7.5, 150 mM NaCl, 1 mM TCEP, 0.005% Tween 20 and for regeneration 1 M sodium chloride solution was used. Interactions of the RXRα LBD with the LBDs of PPARγ WT, T475M and M280I mutants were analyzed in the manner of dose response using twofold dilution series of PPARγ LBDs ranging from 1 to 250 nM. The association phase was 120 s and the dissociation phase was 120 s. After subtracting the reference and buffer signal, the data were fit to a steady state binding model and 1:1 kinetic model to define the $K_D$ and $k_{off}$ using the Biacore T200 Evaluation software (GE Healthcare).

**Mass spectrometry analysis.** Prior to mass spectrometry analysis, PPARγ and all the different mutant proteins were buffer exchanged against 200 mM of ammonium

acetate at pH 6.8, using five cycles of concentration/dilution with a micro-concentrator (Vivaspin, 10-kD cutoff, Sartorius, Göttingen, Germany). All the samples were diluted either in H₂O/ACN/HCOOH (denaturing MS conditions) or in 200 mM AcONH₄ (native MS conditions) to a final concentration of 5 μM and infused with an automated chip based nanoelectrospray device (Triversa Nanomate, Advion Bioscience, Ithaca, USA) operating in the positive ion mode, coupled to a Synapt G2 HDMS mass spectrometer (Waters, Manchester, UK). Under denaturing mass spectrometry conditions, the backing pressure, the cone voltage and the extraction voltage of the mass spectrometer were set to 1.47 mbar, 20 V, and 4 V, respectively. The trap cell was operating under a constant Ar pressure of $9 \times 10^{-3}$ mbar (2 ml/min). The voltage within the trap cell and the trap DC bias were set to 7 V and 2 V, respectively. In this case, the traveling wave-based helium cell and ion mobility cell were used as ion guides under $1.2 \times 10^{-4}$ mbar and $8 \times 10^{-5}$ mbar, respectively. In the transfer cell, ions were transmitted using an acceleration voltage of 7 V under a constant pressure of $10^{-6}$ mbar. The pressure within the time of flight analyzer was kept at $6.4 \times 10^{-7}$ mbar. For native mass spectrometry analysis, the cone voltage, the extraction voltage and the backing pressure of the mass spectrometer were set to 20 V, 5 V, and 6 mbar, respectively, in order to improve the transmission of the "native-like" molecular ions and avoiding ion heating. Ions were efficiently trapped with a constant Ar flow rate of 4.5 ml/min (leading to a final pressure in the trap cell of $8.8 \times 10^{-3}$ mbar) and 4 V and then they were pulse driven with a trap DC bias of 2 V. During mass spectrometry analysis, the pressure within the helium cell and ion mobility cell were kept at $1.2 \times 10^{-4}$ and $8 \times 10^{-5}$ mbar. Ions were transferred from the ion mobility cell to the analyzer by applying a constant voltage of 2 V under a constant pressure of $10^{-6}$ mbar in the transfer cell. Finally, ions were analyzed within the time of flight analyzer under high vacuum conditions ($6.3 \times 10^{-7}$ mbar) in order to ensure highly accurate mass measurements. Mass spectra recorded with Synapt G2 platform were analyzed with MassLynx 4.1 (Waters, Manchester, UK).

**Native IM-MS**. Ion mobility experiments were performed on the Synapt G2 HDMS (Waters, Manchester, UK). The backing pressure, the cone voltage and the extraction cone was set to 6 mbar, 40 V, and 5 V, respectively. The traveling wave-based ion trap was filled with a continuous Ar flow of 6 ml per min and the trap collision energy was set to 4 V. Ions were pulse-driven with a trap DC bias of 60 V and subsequently separated in the TWIMS (Traveling Wave Ion Mobility Spectrometry) cell under a constant N2 pressure of 5.36 mbar (N2 flow rate of 25 ml per min). The IM wave velocity and height were set to 1250 m/s, and 40 V, respectively. The other mass spectrometer parameters not described in this section remained the same as those used during the native mass spectrometry analysis. IM data were calibrated as described elsewhere. Briefly, three charge states of two external calibrants (Cytochrome *C* and β-lactoglobuline) were used to determine the rotationally averaged collision cross section (CCS) of the ion of interest. MassLynx 4.1 was used to perform IM data interpretation. Reported TWCCSN2 values correspond to the average TWCCSN2 measurement performed in triplicate under the same experimental conditions.

**Collision-induced unfolding (CIU) experiments**. Prior to IM separation, ions were progressively activated in the trap cell by increasing the acceleration voltage in 5 V steps from 0 to 60 V. Ion mobility data at each individual voltage were acquired during 1 min, and finally compiled to give rise to the CIU fingerprints. The arrival time distribution (ATD) of each individual charge state of the protein complexes were extracted with the open-source CIU-Suite software[50]. CIU experiments have been performed in triplicate for each individual complex leading to a standard deviation lower than 4%, calculated with the CIUSuite-Stat module included in the CIU-Suite package.

**HDX–MS experiments**. HDX of PPARγ WT, M280I and T475M proteins were carried out with and without PGC1α peptide, with 1:3 concentration ratios in 20 mM Tris (pH 8.0), 200 mM NaCl, 5% glycerol. Samples were incubated for a range of exchange times (0 min, 0.5 min, 2 min, 10 min, 30 and 60 min) in 95% of deuterated buffer (20 mM Tris (pH 8.0), 200 mM NaCl) before quenching the exchange reaction by adding a quench solution (100 mM glycine, 2 M GdHCl, pH 2.2) at 1 °C during 30 s. Protein digestion of quenched samples (120 pmol) was then performed through a pepsin-immobilized cartridge in 0.1% aqueous formic acid solution at a 200 μl/min. The digested peptides were then trapped on UPLC pre-column (ACQUITY UPLC BEH C18 VanGuard pre-column, 2.1 mm I.D. × 5 mm, 1.7 μM particle diameter, Waters) and separated on UPLC column (ACQUITY UPLC BEH C18, 1.0 mm I.D. × 100 mm, 1.7 μM particle diameter, Waters) at 0 °C. Preparation and injection of the samples were automatically conducted by the CTC PAL robot (Leap Technologies, Zwingen, Switzerland), while chromatographic step was carried out on Acquity UPLC system with HDX technology (Waters, Milford, MA, USA). Mass spectrometry measurements were acquired with Synapt G2Si HDMS (Waters) with electrospray ionization, using data-independent acquisition mode (MS^E) over an *m/z* range of 50–2000 and with 100 fmol/μl Glu-FibrinoPeptide as lock-mass correction. HDX experiments were realized in triplicate for each time point. Peptide identification was performed using ProteinLynx Global Server 2.5.3 (Waters) with a custom protein sequence library, where peptide and fragment tolerances

were automatically adjusted by PLGS, while oxidized methionine was set as variable modification. Deuterium uptakes for all identified peptides were checked and validated manually using DynamX 3.0 (Waters): only peptides—with a length range of 5–25 residues—identified in all replicates were kept with a minimum products per amino acid of 0.3. Only one charge state was kept for each peptide and deuterium uptakes were not corrected for back-exchange, representing as relative. HDX–MS results were statistically validated using Mixed-Effects Model for HDX experiments (MEMHDX)[29], using Wald tests, where statistical significance thresholds were set to 0.01.

**Reporting summary**. Further information on experimental design is available in the Nature Research Reporting Summary linked to this article.

## Data availability
Atomic coordinates and related structure factors have been deposited in the Protein Data Bank with accession codes: 6FZY (PPARγ T475M apo), 6FZF (PPARγ T475M-GW1929-PGC1α), 6FZJ (PPARγ M280I-GW1929-PGC1α), 6FZG (PPARγ I290M–GW1929) and 6FZP (PPARγ-GW1929-PGC1α). The source data underlying Figs. 1–3 and Supplementary Figs. 1, 13 and 14 are provided as a Source Data file. A reporting summary for this Article is available as a Supplementary Information file. All other data supporting the findings of this study are available from the corresponding authors on reasonable request.

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

## Acknowledgements

This work was supported by a grant from *Ligue Nationale Contre le Cancer* (C.K., L.C.-T., E.G., R.Z., S.R., Y.N., Y.A., F.R., I.B.-P.) as an associated team (*Equipe labellisée*), the "*Carte d'Identité des Tumeurs*" program initiated, developed and funded by *Ligue Nationale Contre le Cancer*, by a "PL-Bio" project funded by INCa (2016–146), the French Ministry of Education and Research, the CNRS, and the Institut Curie. The project was also supported by the Centre National de la Recherche Scientifique, the Institut National de la Santé et de la Recherche Médicale and the University of Strasbourg. We acknowledge the support and the use of resources of the French Infrastructure for Integrated Structural Biology FRISBI ANR-10-INBS-05, the Instruct-ERIC and the French Proteomic Infrastructure ProFI ANR-10-INBS-08–03. We would like to thank the staff of Proxima 1 at SOLEIL as well as of ID23 and ID30A at ESRF for assistance in using the beamlines. We also thank Alastair McEwen (IGBMC) for help in X-ray data collections and structural refinement and Camille Kostmann (IGBMC) for assistance in SPR. We would like to thank Simone Benhamou (IGR) and Thierry Lebret (Foch Hospital) for their help in setting up the CIT series of bladder tumors. We thank GIS IBiSA and Région Alsace for financial support in purchasing a Synapt G2 HDMS instrument. We acknowledge the Strasbourg University High Performance Computing Center and their staff for providing access to computing resources and for their support. We also acknowledge GENCI (Grand Équipement National de Calcul Intensif) for computing resources and the Equipex Equip@Meso project (Programme Investissements d'Avenir). MB was supported by a fellowship from the Région Alsace. OA-H acknowledges the IdeX program of the University of Strasbourg for funding his postdoctoral fellowship.

## Author contributions

N.R., R.H.S. F.R. and I.B.-P. designed the study. H.L., C.B. and T.M. provided and characterized the tumors from the Strasbourg cohort. C.K., R.Z. and E.G. sequenced PPARG and RXRA and analyzed the data together with L.C.-T., A.K. and A.d.R. who carried out the bioinformatics analysis. A.K., A.d.R. and Y.A. classified the tumors according to pathological status and to the different subgroups identified by the TCGA. C.K., L.C.-T., F.D., S.R., Y.N. and I.B.-P. performed the functional studies and analyzed the data. J.O., W.Z., C.P.-I. and S.H.-B. performed the biochemical, biophysical, and structural studies, and J.O. and N.R. analyzed the data. S.V. and K.A. performed the molecular dynamics, and A.D. and R.H.S. analyzed the MD data. O.A.H. and M.B. performed the mass spectrometry experiments, O.A.H., M.B. and S.C. analyzed the mass spectrometry data. N.R., C.K., L.C.-T., R.H.S., F.R. and I.B.-P. wrote the manuscript. All authors made comments on the manuscript.

## Additional information

**Competing interests:** The authors declare no competing interests.

