## [Peer Review File · Nature Communications]

Reviewers' comments:

Reviewer #1 (Remarks to the Author):

Nat. Commun.

Manuscript ID: NCOMMS-18-08035-T

In this manuscript, the authors identify somatic mutations in the PPARG gene that occur naturally in bladder cancer. These include gain-of-function mutations in the LBD that enhance the transcriptional activity of the receptor, and receptor interaction with coactivator peptides. The authors demonstrated by RNAi that RT4 bladder cancer cells require expression of the endogenous mutant PPAR γ -T475M to proliferate.

The authors also explored the structural basis for the gains-of-function. To this end, the authors solved the X-ray crystal structures of 5 distinct 1.9–3.1 Å mutant PPAR γ LBD complexes and compared these structures to the wild type PPAR γ LBD. These structures suggest that the mutations stabilize an active conformation of the apo receptor LBD. This idea is further supported by molecular dynamics simulations suggesting that the PPAR γ -T475M mutation reduces the local flexibility of the LBD around coactivator binding site.

Major comments

1. Show that the PPAR γ mutations increase coactivator interaction in the context of full-length proteins. This may include a mammalian 2-hybrid assay comparing interactions between transfected wild-type and mutant PPAR γ with NR coactivators, or a chromatin-immunoprecipitation assay comparing recruitment/promoter occupancy of coactivators at PPAR γ /RXR α target genes in cells expressing wild type or mutant PPAR γ .
2. Show that knockdown of wild-type PPAR γ , or another PPAR γ /RXR α mutant, does not reduce proliferation of bladder cancer cells. The authors could try a similar experiment (Fig. 2C) in the HT1997 cell line that harbors an RXR α -S427F mutation (Suppl. Table 3), or any of the remaining 23 cell lines in which they did not detect a PPAR γ mutation?
3. Show that knockdown of PPAR γ -T475M alters the expression of endogenous PPAR γ /RXR α target genes. QPCR examining the mRNA levels of FABP4 and PLIN2 in RT4 cells treated as shown in Fig. 2 should suffice.
4. Supplementary figures 3C and 4 were missing.

Minor comments

Suppl. Table 1: Indicate the sex of the cell lines, if known

Page 5, line 91–93: This sentence is unclear.

Suppl. Table 2: Clarify the column headers. For example, a mutation is a genomic alteration.

Suppl. Tables 4 and 6: One combined table should suffice.

Fig. 2A–B: Use multiple hypothesis testing to infer statistical significance. The Student's T-test is not adequate for multiple hypothesis.

Page 8, line 156–160: Refers to the Suppl. Fig. 2, instead of Suppl. Fig. 1.

Page 8, line 159: "...that that..."

The color scheme and node-types used makes it somewhat difficult for the reader to interpret the data, compared to Fig. 2B.

Suppl. Table 5:

The validation reports suggest that following structures can be improved: 6FZY, 6FZP (side chain outliers), and 6FZG (Ramachandran outliers)

The rationale for analyzing the structures of PPAR γ homodimers should be clearly explained. Clarify the similarity between the homodimer and RXR heterodimer interfaces.

Is there any evidence that mutant PPAR γ can function as a homodimer?

Suppl. Fig. 2: Figure legend (A) The statement... "By contrast to PPAR γ WT apo, in PPAR γ T475M apo, both LBDs forming the homodimer have an agonist" is unclear.

(B) The Met475–Tyr505 interaction is not a "pi-stacking" interaction and should not be described as such.

Suppl. Fig. 3. Showing amino acid side-chains without the peptide linkers (as in Fig. 3C–D) would also help to clarify their relative orientation.

Page 8, line 196: Suppl. Fig. 3C is missing

Page 10, line 207: Suppl. Fig. 4 is missing

Reviewer #3 (Remarks to the Author):

Rochel and colleagues showed activating mutations in PPAR gamma in bladder cancer. They have identified them in a combination of independent cohorts, including publically available data from TCGA. They further investigated the functional consequences of these mutations. They demonstrated that some mutations alter conformation and the structure of the protein. These changes could be potentially be targets for pharmacological development. They also emphasize that these data highlights the idea of PPAR gamma-dependency in bladder cancer.

- 1- Why did the authors decide to collect only one point (96h after transfection) for the knockdown- RNAi experiment? A growth curve with points from 24-96h could show the reader the dynamic of growth due to this inhibition.
- 2- For the overexpression, why did the authors choose the 48h point? Did they test other time points?
- 3- Please add IRB/ethics committee authorization information.
- 4- The results could be sub divided into sections to be easier to follow and to compare to the figures.
- 5- The authors should clearly state that they have sequenced using conventional sequencing and looked at deep sequencing data publically available for comparison, and then grouped both results and considered all as a group. The two methodologies have different sensitivities and this could interfere with the mutation detection. The two groups should be considered as separate groups.

Point by point answer to reviewers' comments

We thank the reviewers for their comments, which helped us to improve our manuscript.

Reviewer #1 (Remarks to the Author):

Manuscript ID: NCOMMS-18-08035-T

In this manuscript, the authors identify somatic mutations in the PPARG gene that occur naturally in bladder cancer. These include gain-of-function mutations in the LBD that enhance the transcriptional activity of the receptor, and receptor interaction with coactivator peptides. The authors demonstrated by RNAi that RT4 bladder cancer cells require expression of the endogenous mutant PPAR γ -T475M to proliferate. The authors also explored the structural basis for the gains-of-function. To this end, the authors solved the X-ray crystal structures of 5 distinct 1.9–3.1 Å mutant PPAR γ LBD complexes and compared these structures to the wild type PPAR γ LBD. These structures suggest that the mutations stabilize an active conformation of the apo receptor LBD. This idea is further supported by molecular dynamics simulations suggesting that the PPAR γ -T475M mutation reduces the local flexibility of the LBD around coactivator binding site

Major comments

1. Show that the PPAR γ mutations increase coactivator interaction in the context of full-length proteins. This may include a mammalian 2-hybrid assay comparing interactions between transfected wild-type and mutant PPAR γ with NR coactivators, or a chromatin-immunoprecipitation assay comparing recruitment/promoter occupancy of coactivators at PPAR γ /RXR α target genes in cells expressing wild type or mutant PPAR γ .

We thank the reviewer for this comment. We have now evaluated the interaction of full-length PPARG proteins (wild-type and mutants) with MED1 co-activator peptide using a mammalian 2-hybrid assay in HEK293FT cells (Fig.3C). Our new results confirmed our previous data obtained using the LBD domain of PPARG (wild-type and mutants) showing that PPARG mutations increased co-activator interaction (Fig. 3B).

2. Show that knockdown of wild-type PPAR γ , or another PPAR γ /RXR α mutant, does not reduce proliferation of bladder cancer cells. The authors could try a similar experiment

(Fig. 2C) in the HT1997 cell line that harbors an RXR α -S427F mutation (Suppl. Table 3), or any of the remaining 23 cell lines in which they did not detect a PPAR γ mutation?

We have shown here that PPARG-T475M mutant depletion reduced the proliferation of RT4 bladder cancer cells. We had previously shown that the overexpression of wild-type PPARG as a consequence of PPARG amplification renders bladder tumor cell lines PPARG-dependent whereas cells presenting low expression level of PPARG were not sensitive to PPARG depletion (Biton et al., Cell Reports, 2014, PMID: 25456126). These results were confirmed by a study by Goldstein et al. showing that cell lines presenting a PPARG-activation signature were sensitive to pharmacological inhibition of PPARG. These PPARG-activated cell lines included HT1197 cells expressing RXRA-S427F (Goldstein et al., Cancer research 2017, PMID: 28923856). Two other studies have also demonstrated that RXRA-S427F induces, in a ligand-independent way, PPAR transcriptional signaling (Hastead et al., elife 2017, PMID: 29143738; Korpál et al., Nature Communications 2017, PMID: 28740126). Taken together, these experiments suggest that different genetic alterations can lead to the activation of a PPAR γ /RXR α pathway which render bladder cancer cells PPARG-dependent. We have developed more this point in the Results section to make our point clearer.

3. Show that knockdown of PPARG-T475M alters the expression of endogenous PPAR γ /RXR α target genes. QPCR examining the mRNA levels of FABP4 and PLIN2 in RT4 cells treated as shown in Fig. 2 should suffice.

We thank the reviewer for this comment. We have now evaluated the effect of PPARG-T475M depletion on the expression of three endogenous PPAR γ /RXR α target genes in RT4 cells. PPARG-T475M depletion inhibited FABP4, PLIN2 and ACSL5 expression (Fig. 2C, right panel).

4. Supplementary figures 3C and 4 were missing.

We apologize for this. Supplementary figures have been included in the revised version of the manuscript.

Minor comments

Suppl. Table 1: Indicate the sex of the cell lines, if known

Sex of the cell lines, if known, have been added to the Table

Page 5, line 91–93: This sentence is unclear.

The text has been modified.

Suppl. Table 2: Clarify the column headers. For example, a mutation is a genomic alteration.

We modified the column header to “PPARG copy number alteration”.

Suppl. Tables 4 and 6: One combined table should suffice.

For a better distinction between mutations found in bladder and other cancers types and mutations found in cancer types other than bladder cancers, we prefer to keep two separate Supplementary Tables.

Fig. 2A–B: Use multiple hypothesis testing to infer statistical significance. The Student’s T-test is not adequate for multiple hypothesis.

In response to the reviewer’s request, we have now used the Dunnett's multiple comparisons test to infer statistical significance. This test should be adequate for multiple comparisons.

Page 8, line 156–160: Refers to the Suppl. Fig. 2, instead of Suppl. Fig. 1.

Page 8, line 159: “...that that...”

These points have been noted and the text modified accordingly.

The color scheme and node-types used makes it somewhat difficult for the reader to interpret the data, compared to Fig. 2B.

The color and node-types of the figure has been changed.

Suppl. Table 5: The validation reports suggest that following structures can be improved: 6FZY, 6FZP (side chain outliers), and 6FZG (Ramachandran outliers) The rationale for analyzing the structures of PPARg homodimers should be clearly explained. Clarify the similarity between the homodimer and RXR heterodimer interfaces. Is there any evidence that mutant PPARg can function as a homodimer?

We agree with the reviewer that the structured could be improved. All structures were further refined and side chain and Ramachandran outliers corrected when possible. PPARg is

monomeric in solution and crystalized both as a monomer and very often as a homodimer with the canonical dimer interface observed in the heterodimer formed with RXR. There is no evidence that PPAR γ functions as a homodimer *in vivo*, however the same homodimer interface is also observed in the estrogen receptor ER and RXR homodimers.

Suppl. Fig. 2: Figure legend (A) The statement... “By contrast to PPAR γ WT apo, in PPAR γ T475M apo, both LBDs forming the homodimer have an agonist” is unclear.

We have clarified this statement in the text: “Of these structures, the apo protein and some complexes with partial agonists display one monomer in an active conformation and one monomer in an inactive conformation with a different positioning of helix 12. Helix 12 is key regulatory structural element in the activation function by interacting with coregulators. These 2 conformations are generally designated as fully active and inactive.”

(B) The Met475–Tyr505 interaction is not a “pi-stacking” interaction and should not be described as such.

We corrected and removed reference to pi-stacking.

Suppl. Fig. 3. Showing amino acid side-chains without the peptide linkers (as in Fig. 3C–D) would also help to clarify their relative orientation.

This modification has been done.

Page 8, line 196: Suppl. Fig. 3C is missing

Page 10, line 207: Suppl. Fig. 4 is missing

We apologize for this. Supplementary figures have now been added in the revised version of the manuscript.

Reviewer #2 (Remarks to the Author):

In the manuscript by Rochel et al., entitled, “Recurrent activating mutations of PPAR γ -associated with luminal bladder tumors,” the authors identify activating mutations of PPARG and attempt to explain three of the mutations by structural methods. Mutations were identified by means of 359 bladder tumours and the CGA atlas. 8 of the mutations were investigated by transactivation data in the PPRE system without ligand addition (although the authors say they compared to rosi but did not include). Ultracentrifugation was used to ensure PPAR-RXR

heterodimer still formed. Crystal structures of wild type LBDs as well as three mutants were formed. A cursory explanation was given as to the structural mechanism of activation. Given that PPAR has been found to play a role in many cancers (and antagonists can be potentially used as treatment) and there are problems with some of the data, I suggest this paper is better suited for a more specialized journal.

Problems include:

The title appears to have a typo. It should be PPAR associated or PPAR-associated.

This point has been taken into consideration and the text modified accordingly.

The abstract was poorly written and difficult to understand. Terms like “transcription program” and “focal gains” are confusing to the reader (and reviewer).

This point has been taken into consideration and the text modified.

The introduction is only one page. This does not seem to give enough background knowledge to the reader to understand the context of the entire paper.

The introduction has been more developed to bring more details about the background of the paper.

Is detecting 3.4% PPAR mutations in tumours of significance?

We agree that the mutation rate of PPARG is low, but the existence of these mutations brings another genetic argument to back up the importance of the PPARG/RXRA pathway in luminal bladder tumors. Considering PPARG amplifications, RXRA and PPARG mutations, genetic alterations of the PPARG/RXRA pathway can be observed in 25-30 % of the luminal bladder tumors. In addition, the identification of these mutations in human tumors allows studying structure/activity relationship of an important pro-tumorigenic protein in luminal bladder tumors.

Sentence 91 says isoform 1 is only expressed in adipose cells but is also expressed ubiquitously. Can't be both can it?

We apologize about this typo mistake. The text has been modified to reflect the fact that isoform 2 is only expressed in adipose cells, while isoform 1 is ubiquitously expressed.

The PPRE system is known for having low fold upregulation in response to ligands. Additionally, basal level transcription of PPAR is known to be rather low. Why was the rosi data not included, it should have, and in fact better than rosi would have been an endogenous ligand. It would have also been better to use the GAL4 system for transactivation to better observe differences in transcription as the fold response is much better.

We agree that the PPRE system has some limitations. That is why we used two complementary approaches to evaluate the transcriptional activity of the different mutants: the PPRE system in HEK293FT and the analysis of PPARG endogenous target genes expression after PPARG mutants overexpression in 5637 bladder cancer cells. We initially used the PPRE system in U2OS cells but, due to a better transfection efficiency, we used in the revised version of the manuscript the PPRE system in HEK293FT cells. We have added these results in presence of rosiglitazone for the two approaches (Supplementary Fig.1A and 1D). Also, for the mutant PPARG-T475M, we have compared our results with the PPRE system to results with the GAL4 system and we observed similar results (see Figure for reviewer).

Binding is not a binary effect. The ultracentrifugation experiments are of only minimal value. Given how close the T475 mutant is to the RXR interface Kd's of PPAR for RXR should have been calculated.

We included the ultracentrifugation experiments to show the oligomeric state of the PPAR WT and mutants in solution and their ability to form heterodimer with RXR. To provide the Kd of PPAR-RXR interaction, we have now included SPR data (Figure 3A and Supplementary Fig. 4).

All PPARg structures to date are in an agonist conformation. This makes claims in lines 188-191 dubious. While it is clear that T475M is making contact with the Tyr it also is on the RXR interface. The authors have not adequately shown the mechanism of action of this mutation.

Due to space limit of the manuscript, initially submitted to another of Nature's journals that was then transferred to Nature Communications, the description of the structural analysis and molecular dynamics results for T475M mutant was rather short. We agree that the mechanism of action of this mutation was not fully described. To clarify this point, we have now included the structural analysis of the effect of the mutation on heterodimer stability (Fig. 3A and Supplementary Fig.4). However, concerning the reviewer's comment on the agonist conformation of all PPARg structures, we do not agree. Indeed, among the structures of PPARg deposited in the PDB, there is the apo protein (1PRG) and several complexes with partial agonists (2Q6R, 2I4Z, 3WMH) that exhibit one monomer in an active conformation and the other monomer in an inactive conformation as distinguished by a different positioning of helix 12. These two conformations were designated as fully active and inactive. Although these two conformations of helix 12 in the crystal structures are influenced by crystal packing, they reflect the dynamic character of helix 12 in absence or in presence of a partial agonist ligand that was further confirmed by solution structural methods (Chrisman I.M. et al. Nat Comm 2018,9:1794).

The authors provided cif file structure factors making it the absolutely most difficult way for reviewers to access to electron density. It's almost as if they purposely tried to make it difficult for the reviewers to make maps. Map files or map coeffs in mtz files would have been nice given they were asked for specifically. Additionally, I could not find the data for the I290M mutation structure.

The cif files for all mutants were provided as requested by the editor and correspond to the validated files by the PDB that can be download from PDB when released. We now have provided map files.

The structures don't seem to be finished as noted by large amounts of difference density left in the models.

The structures were refined with good quality as assessed by the quality scores when compared to structures of similar resolution notably of PPARg complexes deposited in the PDB. However, in response to the reviewer's comments, all structures were further refined by adding missing waters and double side chain conformations. Some regions remain more flexible (higher temperature factor, poor side chain density). For most of the structures refined, the quality scores were further improved.

The residues in the paper and the residues in the structure use different isoform numbering making it very difficult for the reviewers/authors to make sense of anything.

We agree that the numbering was confusing. For clarity, all structures files have now been numbered using PPARg2 isoform.

The data table should contain CC1/2 and Rpim.

These data have been included in the RX data table.

For a structure trying to detect small differences in structures, a 3.1 resolution structure is not high enough resolution to publish.

This structure was included to describe the conformations of helix 12 in the two monomers of T475M mutant compared to WT protein.

The pdb codes should be put in the data table.

As requested by the journal, the pdb codes are listed in the Methods and were included in the data table.

There should be stereo images of reduced model biased maps somewhere highlighting the important section (ie the mutation) of each structure, probably in the supplement.

We thank the reviewer for this remark. We have now included a supplementary fig. 11A showing a stereo view of the mutated residues in unbiased omit Polder maps with reduced model bias and exclusion of solvent density.

Supp figure 3 lacks labels.

We apologize for this. Labels have been added.

The explanation on page 11 of how the H2' mutation is activating is totally unbelievable with the data provided. More experimental evidence is needed to justify that.

We thank the reviewer for this remark and agree that the description of the structural analysis of the effect of the mutants was very short. This is particularly true for M280 and I290 mutants. We now have described in more details these two structures and their dynamics. We have also included new data on HDX-MS that provide complementary information on the effect of these two mutants on H3 stabilization (Supplementary Fig. 12-13).

Concerning is that the structure containing the M280I mutation doesn't seem to fit the density very well. I bet a composite omit map might even look worse.

The polder omit map of M280I clearly shows its correct positioning. But it is true that Ω loop is very flexible and often poorly defined in the electron density maps of the published structures of PPAR γ .

Reviewer #3 (Remarks to the Author):

Rochel and colleagues showed activating mutations in PPAR gamma in bladder cancer. They have identified them in a combination of independent cohorts, including publically available data from TCGA. They further investigated the functional consequences of these mutations. They demonstrated that some mutations alter conformation and the structure of the protein. These changes could be potentially targets for pharmacological development. They also emphasize that these data highlights the idea of PPAR gamma-dependency in bladder cancer.

1- Why did the authors decide to collect only one point (96h after transfection) for the knockdown- RNAi experiment? A growth curve with points from 24-96h could show the reader the dynamic of growth due to this inhibition.

We initially chose 96h after transfection to evaluate the effect of PPARG depletion on cell viability based on our past experience evaluating the effect of PPARG depletion on cell viability of cells presenting PPARG amplification (Biton et al., Cell Reports 2014, PMID: 25456126). In response to the reviewer's question, we have now evaluated the effect of PPARG depletion on cell viability of RT4 cells from 24h to 96 h after siRNA transfection. We observed a significant reduction of cell viability from 72 h. The manuscript has been modified to reflect this data (Fig. 2C, left panel).

2- For the overexpression, why did the authors choose the 48h point? Did they test other time points?

For the reporter assay in HEK293FT cells, we tested 24h, 48h and 72h. Significant results were obtained from 48h after the overexpression of the PPARG wt and mutants proteins. So, based on these experiments, we chose this time point for the overexpression in 5637 cells and the analysis of PPARG target genes expression

3- Please add IRB/ethics committee authorization information.

We have now added IRB/ethics committee authorization information in the Material and Methods section of the manuscript.

4- The results could be sub divided into sections to be easier to follow and to compare to the figures.

We have now divided the results into sections corresponding to each Figure.

5- The authors should clearly state that they have sequenced using conventional sequencing and looked at deep sequencing data publically available for comparison, and then grouped both results and considered all as a group. The two methodologies have different sensitivities and this could interfere with the mutation detection. The two groups should be considered as separate groups.

We have now more clearly described the sequencing methods used for the different tumors and considered two separate groups according to the two different methodologies used.

Reviewer #1 (Remarks to the Author):

The authors have done a very thorough job in addressing the reviewers concerns, including providing the requested data, statistical approaches, and analyses.

The authors should work with the editors on the abstract to improve readability. I was put off by the first sentence and had to reread it a couple of times. Perhaps something like: " A key event in luminal bladder tumors is amplification of PPAR γ leading to the activation of PPAR γ /RXR transcriptional activity. This renders the tumor dependent on PPAR γ for growth and modulates the tumor microenvironment to favor escape from immunosurveillance." I also don't understand what it means to say that mutations converge to activate. This makes is sound like there are multiple mutations in the same patient. Is this what you meant? Might be less confusing to just say they activate.

Overall the revision was very readable and a very nice story!

Kendall Nettles

Reviewer #2 (Remarks to the Author):

The authors have now submitted a draft with significantly more data and answered many of the questions. The major point the authors are making in this manuscript is that the mutations they have discovered in certain cancers within PPAR are activating and, more importantly, their analysis defines the structural mechanism. Their ultimate claim is that the mutations stabilise the "active" conformation allowing for more co-activator recruitment and hence higher than normal transactivation. I do not find these conclusions warranted for the reasons listed below. Papers like this have been published before, and due to lack of novelty as described here, have been published in lower impact journals (see PMID:25004973).

1. Multiple mutations were identified, yet only a handful of them were analysed due to convenience making the case weak, why study only half of them? Full-length structures can be elucidated for PPAR now.
2. Transactivation rates were performed in HEK cells. This is hardly physiological and may not have the appropriate co-regulators in the appropriate ratios to make real conclusions.
3. RNA levels of the 3 selected genes in appropriate cells were not overly convincing due to the degree of error in the mutant samples. Also, strange is the use of only 3 genes being studied.

4. SPR was carried out. RXR can exist as tetramers, dimers, and monomers. Hard to know what is actually applied on the chip. If a tetramer is covalently attached to the chip how can PPAR heterodimerise with RXR. This is basically adding 3 ligands (RXR monomer, dimer, and tetramer) and two analytes (PPAR monomer and dimer). Additionally, only one data set was reported with no error. Even so, none of this shows the actual structural mechanism; just that RXR Kd with PPAR doesn't change. Same with MS and UC.

5. Figure 3 C is not believable due to large error in the mutant samples. Even if it was, it would not decipher the structural mechanism.

6. All apo crystal structures (and some others) solved in space group C2 have a crystal contact artefact. This is well known in the field. Their description of this in the text is disingenuous. None of their space groups are C2, and most mutants are forced into an active state either by addition of co-activator or agonists. Saying that the mutants stabilize an agonist conformation after being forced into an agonist conformation in the structure by adding co-activators or agonists proves nothing. It just allows for speculation that needs to be backed up by additional experiments. Of the dozens of structures of the PPAR LBD only 1-2 are not in an agonist locked conformation, making the ability to draw conclusions from their structures dubious. This is a logical fallacy called begging the question.

7. The HDX shows stabilization in many areas not just in the area they speculate as activating. Therefore, this is also unconvincing of any model they propose.

8. "The mutated residue T475M interacts with the C-terminal Y505 (Fig.4A) stabilising helix 12 into the active conformation and leading to a more stable interaction with the co-activator peptide." What is the structural claim for this? Why do the authors not show the wild type interaction of the Tyr? It forms many h-bonds of low distance and favourable energy. Why are we to believe the hydrophobic interactions are better and more stabilizing? What evidence is there that the hydrophobic interactions packing against the Tyr are more stable for the co-regulator binding?

Reviewer #3 (Remarks to the Author):

None

Point by point answer to reviewers' comments

We thank the reviewers for their comments, which helped us to further improve our manuscript.

Reviewer #1 (Remarks to the Author):

The authors have done a very thorough job in addressing the reviewers concerns, including providing the requested data, statistical approaches, and analyses.

The authors should work with the editors on the abstract to improve readability. I was put off by the first sentence and had to reread it a couple of times. Perhaps something like: " A key event in luminal bladder tumors is amplification of PPAR γ leading to the activation of PPAR γ /RXR transcriptional activity. This renders the tumor dependent on PPAR γ for growth and modulates the tumor microenvironment to favor escape from immunosurveillance." I also don't understand what it means to say that mutations converge to activate. This makes is sound like there are multiple mutations in the same patient. Is this what you meant? Might be less confusing to just say they activate.

Overall the revision was very readable and a very nice story!

Kendall Nettles

We thank Dr. Nettles for his interest on our study and for his comments that lead to an improved readability of the abstract. We have now proposed a modified version of the abstract accordingly.

Reviewer #2 (Remarks to the Author):

The authors have now submitted a draft with significantly more data and answered many of the questions. The major point the authors are making in this manuscript is that the mutations they have discovered in certain cancers within PPAR are activating and, more importantly, their analysis defines the structural mechanism. Their ultimate claim is that the mutations stabilise the "active" conformation allowing for more co-activator recruitment and hence higher than normal transactivation. I do not find these conclusions warranted for the reasons listed below. Papers like this have been published before, and due to lack of novelty as described here, have been published in lower impact journals (see PMID:25004973).

With all due respect, we disagree with the reviewer's comments about the novelty and significance of our results. In this work, we identified for the first time recurrent mutations of PPAR γ in cancer. We showed for the first time that a PPAR γ mutant displays pro-tumorigenic activities in bladder tumors. We demonstrated for the first time that PPAR γ mutations could confer gain of function and we identified functional mechanisms that underlay these gains of function for three mutants that show a better affinity for both RXRA co-receptor and co-activators such as MED1 and PGC1A. And finally we defined structural mechanisms that underlay these gains of function for these three mutants.

We therefore believe that the paper we submit here contains significantly more novel results than the one referred to by the reviewer, which presents the structural study of a single known loss of function mutant of PPAR γ related to lipodystrophy (PMID:25004973).

1. Multiple mutations were identified, yet only a handful of them were analyzed due to convenience making the case weak, why study only half of them? Full-length structures can be elucidated for PPAR now.

As stated in the text, 7 out of 8 recurrent mutations were analyzed in functional assays in both 5637 and HEK293FT cells. We decided to further investigate the recurrent mutations showing the highest activity, which correspond to 3 mutations in the ligand binding domain among which is the mutation T475M, the most frequent recurrent mutation. As the reviewer pointed out, full-length structures of PPAR can be elucidated. However, the only reported structures are at resolutions above 3.1 Å, a resolution that reviewer 2 indicated in his first review is “not high enough resolution to publish” (we have included in our manuscript an apo structure at 3.1 Å showing only the overall conformation without going into structural details as suggested by the reviewer)

2. Transactivation rates were performed in HEK cells. This is hardly physiological and may not have the appropriate co-regulators in the appropriate ratios to make real conclusions.

Transactivation rates were initially performed in U2OS cells providing consistent results with those observed in HEK293 cells, which suggests a robustness of the transactivation assay independently of the cell context. During the revision process of the manuscript, we noticed a better transfection efficiency coupled to a higher luciferase signal in HEK293 cells and therefore presented these data in the revised version of the manuscript. We agree that transactivation assay in a non-bladder cancer cell line could not be totally physiological due to the choice of a PPARG promoter motif, as well as to inappropriate co-regulators. That’s why we confirmed our results in another, more physiological context, measuring the endogenous expression of known PPARG target genes after mutant expression in 5637 cells that express low level of endogenous PPARG.

3. RNA levels of the 3 selected genes in appropriate cells were not overly convincing due to the degree of error in the mutant samples. Also, strange is the use of only 3 genes being studied.

We agree that the expression levels of target genes observed in each independent transfection experiment of PPARG (wild-type and mutant) are different leading to a certain degree of error, but the differences are still significant when using a robust statistical test with correction for multiple comparisons. Moreover, it is noteworthy that in each individual experiment, the activity of the mutants was higher than the one of the wild type. We have now showed dot plots for each individual experiment on the graph for a better highlight of the results.

We selected only three genes to evaluate the activity of the mutants since these genes were well known to be target genes of PPARG. The intention of this work was not to evaluate the pattern of genes regulated by PPARG wild-type and mutants, but to provide a proof of concept that mutants did present a higher transactivation activity in a physiological context. For more confidence on the selection of our three readout genes, we combined analysis of our published transcriptomic data after PPARG knock-down using siRNA in SD48 cells presenting a gain of PPARG (Figure A for reviewer) (Biton et al., Cell Rep 2014, PMID: 25456126) and of our PPARG ChIP-Sequencing data (unpublished results) (Figure B for reviewer). This analysis validated that these three genes were indeed directly regulated by PPARG in a luminal bladder cancer cell line. Such a strategy, assessing

PPARG signaling activity by measuring the expression of such a panel of PPARG target genes, has also been used by Korpál et al., (Nature communication 2017, PMID: 28740126) and Halstead et al., (Elife 2017, PMID: 29143738) to demonstrate the activation of PPARG pathway by RXRA mutants.

4. SPR was carried out. RXR can exist as tetramers, dimers, and monomers. Hard to know what is actually applied on the chip. If a tetramer is covalently attached to the chip how can PPAR heterodimerise with RXR. This is basically adding 3 ligands (RXR monomer, dimer, and tetramer) and two analytes (PPAR monomer and dimer). Additionally, only one data set was reported with no error. Even so, none of this shows the actual structural mechanism; just that RXR Kd with PPAR doesn't change. Same with MS and UC.

We mentioned in both the Results and Methods parts that, for biophysical characterization of PPAR-RXR dimer, we used monomeric RXR. To provide justification for these statements, we have now included the gel filtration profiles showing monomeric RXR and monomeric PPAR. Also, the

monomeric states of RXR and PPAR were clearly shown in analytical ultracentrifugation (the sedimentation coefficients obtained correspond to the mass of the monomeric species). The native MS also shows the monomeric states of both proteins. AUC and native gels show that, upon mixing stoichiometric amount of PPAR and RXR, only one species corresponding to the heterodimer is observed. For the SPR measurements, monomeric RXR was attached to the chip and monomeric PPAR, the only form in solution, was used as the analyte. Furthermore, we compared the interactions of monomeric PPAR mutants to monomeric PPAR wild-type, with similarly attached RXR. We showed that, while the Kds were similar for M280I mutant, they changed for T475M mutant, thus providing important information to support the proposed structural mechanism. Figure 3A is showing representative data set used for SPR analysis of the interactions with the corresponding fitting (dashed line) and the chi2 value, while data analysis by 1:1 kinetic model and mean kinetic parameters and equilibrium dissociation constants of at least two independent sets of kinetic experiments are reported in Supplementary Figure 5.

The structural elements explaining the increased activity of the mutants were deciphered through the combination of the various methods used in this work.

5. Figure 3 C is not believable due to large error in the mutant samples. Even if it was, it would not decipher the structural mechanism.

We initially presented the results for three independent experiments by calculating for each mutant a mean relative RLU as compared to the wild-type PPARG for each experiment performed in quadruplicates. This integration of the biological replicates leads to an important error since transfection efficiency could indeed vary significantly between experiments. It is therefore more often shown results for a representative experiment. We agree that the representation of our results could lead to an underestimation of the effects of the mutations, but each individual experiment showed a significant increased binding of MED1 by mutants as compared to wild-type. We now provide such a representative result for one experiment conducted in quadruplicates in Figure 3C.

These experiments did not aim at deciphering the structural mechanism, but the functional mechanism that leads to an increased transcriptional activity of the mutants. Our results showed a better affinity of the mutants for the co-activators that could account for, at least partly, the higher transactivation activity.

6. All apo crystal structures (and some others) solved in space group C2 have a crystal contact artefact. This is well known in the field. Their description of this in the text is disingenuous. None of their space groups are C2, and most mutants are forced into an active state either by addition of co-activator or agonists. Saying that the mutants stabilize an agonist conformation after being forced into an agonist conformation in the structure by adding co-activators or agonists proves nothing. It just allows for speculation that needs to be backed up by additional experiments. Of the dozens of structures of the PPAR LBD only 1-2 are not in an agonist locked conformation, making the ability to draw conclusions from their structures dubious. This is a logical fallacy called begging the question.

We clearly state in our manuscript that “helix 12 conformations observed in the crystal structures are influenced to some degree by crystal packing”. We mention this only for our apo PPAR T457M mutant structure as compared to the 3 other apo structures described in the literature for WT PPAR (Nolte et al., Nature 1998; Uppenberg et al., JBC 1998; Ohashi et al., Bioorg Med Chem Lett. 2015).

We further mention that both monomers are in active conformations, which is in contrast to the WT structures.

The stabilization of helix 12 observed in the holo crystal structures of the mutants, which crystallized in P222, P121 and P422 space groups, was backed up by data from additional experiments including molecular dynamics simulations and HDX-MS.

7. The HDX shows stabilization in many areas not just in the area they speculate as activating. Therefore, this is also unconvincing of any model they propose.

The crystal structures, as well as the molecular dynamics simulations, clearly show (Figure 5B) that in agreement with the HDX data, several regions of the LBD show less flexibility. In the structural analysis, we emphasized the region around the mutation and the coactivator binding site bringing forth the structural stabilization that can explain the functional transcriptional data.

8. “The mutated residue T475M interacts with the C-terminal Y505 (Fig.4A) stabilising helix 12 into the active conformation and leading to a more stable interaction with the co-activator peptide.” What is the structural claim for this? Why do the authors not show the wild type interaction of the Tyr? It forms many h-bonds of low distance and favourable energy. Why are we to believe the hydrophobic interactions are better and more stabilizing? What evidence is there that the hydrophobic interactions packing against the Tyr are more stable for the co-regulator binding?

The crystal structure of T475M mutant reveals that the methionine interacts with Y505 (Figure 4A) while in the wild-type protein T475 does not interact with Y505 (shown in Supplementary Figure10A-B). In addition, the molecular dynamics simulations, which started from the wild-type protein structure and into which the mutation M475 was built based on the wild type T475 conformation, shows the formation of this M475/Y505 interaction. This additional interaction, together with the electrostatic interactions formed by Y505 with E352 and R425 leads to stabilization of H12, which has been shown to be important for cofactor binding.

Furthermore, the molecular dynamics simulations (Figure 4B and Supplementary Figures 9 and 10C comparing T475M and WT) and the subsequent free energy calculations clearly show that a more stable interaction of the coactivator is observed for the T475M mutant.

Concerning the reviewer’s comment about the stabilization associated with hydrophobic “interactions”, many examples exist in the literature pointing to the stabilizing effects of introducing hydrophobic amino acids, dating from an early study (van den Burg et al., Eur. J. Biochem 1994. 220, 981-985) to a recent study (De Taeye SW, et al, J Biol Chem. 2018 293, 1688-1701). In addition, the molecular dynamics simulations clearly show the structural stabilization of H12 upon mutation to M475 (Figure 4B) and a more favorable binding free energy for the coactivator peptide, as described in the text. In addition, we returned to the molecular dynamics simulations to calculate the van der Waals interaction and we found that the van der Waals interaction energy between residue 475 and Y505 increased by about a factor of 10 for M475 with respect to T475. Altogether, the simulation results support the proposed mechanism that the structural stabilization of H12 can result in structural stabilization of the coactivator peptide leading to a better binding.

REVIEWERS' COMMENTS:

Reviewer #1 (Remarks to the Author):

This manuscript should be of broad interest due to the clinical relevance of gain of function mutants of PPAR γ in bladder cancer, the mechanistic studies, and use of multiple structural approaches to characterize the mutants, including x-ray crystallography, HDX, and molecular dynamics simulations. i think the paper was very interesting in revealing mechanisms of allosteric signaling. The authors made significant improvements to the manuscript. The identification of mutants in the nuclear receptor superfamily that cause cancer has been a hot topic, including the androgen and estrogen receptors as well.